# Zero-field quantum beats and spin decoherence mechanisms in CsPbBr₃ perovskite nanocrystals

Rui Cai[1], Indrajit Wadgaonkar[1], Jia Wei Melvin Lim [1,2], Stefano Dal Forno[1], David Giovanni[1], Minjun Feng [1], Senyun Ye[1], Marco Battiato [1] ✉ & Tze Chien Sum [1] ✉

Coherent optical manipulation of exciton states provides a fascinating approach for quantum gating and ultrafast switching. However, their coherence time for incumbent semiconductors is highly susceptible to thermal decoherence and inhomogeneous broadening effects. Here, we uncover zero-field exciton quantum beating and anomalous temperature dependence of the exciton spin lifetimes in CsPbBr₃ perovskite nanocrystals (NCs) ensembles. The quantum beating between two exciton fine-structure splitting (FSS) levels enables coherent ultrafast optical control of the excitonic degree of freedom. From the anomalous temperature dependence, we identify and fully parametrize all the regimes of exciton spin depolarization, finding that approaching room temperature, it is dominated by a motional narrowing process governed by the exciton multilevel coherence. Importantly, our results present an unambiguous full physical picture of the complex interplay of the underlying spin decoherence mechanisms. These intrinsic exciton FSS states in perovskite NCs present fresh opportunities for spin-based photonic quantum technologies.

Semiconducting nanostructures are promising platforms for logic operations and optical modulation with controllable carrier and exciton states[1–4]. For instance, in single quantum dots (QDs), the orthogonal photoluminescence (PL) doublets originating from the superposition of spin up and down states, can serve as a basis set for quantum computing and optical gating[5]. Such coherent optical control can be achieved by leveraging the exciton fine-structure splitting (FSS) in confinement-enhanced semiconductor QDs[6,7], valley excitons in transition metal dichalcogenides (TMDs)[8], and the exciton complexes in low dimensional structures[9–11]. Nevertheless, strong decoherence is usually triggered by increasing temperature or intense excitation because of the enhanced exciton-phonon and/or exciton-exciton Coulomb interaction[12–14]. Furthermore, the small energy splitting between two interacting FSS levels is highly prone to thermal energy and inhomogeneous broadening effects. This typically limits the

observation of such exciton coherence to extremely low temperatures and at the single-particle (nanostructure) level. Therefore, for practical optospintronic applications, it is imperative to search for new semiconductor systems that could transcend these limitations.

Lead halide perovskites (LHPs) and their nanostructures are intriguing candidates for functional devices exhibiting outstanding optoelectronic properties[15] and rich optical spin physics[16–23]. Interestingly, recent single-particle PL measurements reveal the FSS of exciton states in CsPbX₃ NCs because of the strong spin-orbit coupling (SOC) and shape anisotropy in the absence of strong quantum confinement[24–28]. In CsPbBr₃ NCs, the Rashba effect flips the dark singlet and bright triplet states yielding fast radiative recombination of excitons[25], upon which coherent single-photon emission has also been evidenced in single CsPbBr₃ NCs[29]. The existence of exciton FSS in the family of facile processable, defect tolerant, bandgap-tunable colloidal

[1]Division of Physics and Applied Physics, School of Physical and Mathematical Sciences, Nanyang Technological University, Singapore, Singapore. [2]ERI@N, Interdisciplinary Graduate School, Nanyang Technological University, Singapore, Singapore. ✉e-mail: Marco.battiato@ntu.edu.sg; Tzechien@ntu.edu.sg

perovskite NCs, provides a highly promising system for coherent optical manipulation of the excitonic degree of freedom. However, these non-resonant steady-state PL studies do not reveal the coherence dynamics. There is also a lack of time-resolved PL studies on exciton coherence in perovskites probably because of the fast exciton recombination and resolution limits of the technique. Presently, the spin decoherence mechanisms are under intense debate. Strong SOC induced spin relaxation has been empirically suggested to originate from the Elliott-Yafet (EY) mechanism[30,31] in $CH_3NH_3PbI_3$ thin films[16] and $CsPbI_3$ NCs[32]; while it is the D'yakonov-Perel' (DP) mechanism[33] for $CsPbBr_3$ thin films[34] and layered perovskites[35]; and the polaron mediated Maialle-Silva-Sham (MSS) mechanism[36] in $CsPbBr_3$ quantum wells[37]. These mechanisms could possibly be modified by the FSS due to the exchange interaction.

Herein, we explicate the coherent dynamics of the exciton FSS states in colloidal perovskite NC ensembles using circularly polarized transient absorption (CTA) spectroscopy. Upon optical orientation, the exciton multilevel coherence remains preserved and manifests as an oscillatory spin dynamics signature at low temperatures, which originates from the quantum beating between two FSS states. Very interestingly, these coherent dynamics are relatively unaffected by the inhomogeneous broadening caused by the NC size distribution. While at high temperatures with strong interactions with phonons, we find EY process to dominate the spin decoherence, at lower temperatures a motional narrowing process controls the spin dynamics. Such motional narrowing is observed to evolve through two regimes (strong and weak scattering) producing an anomalous temperature dependence of exciton spin lifetime, which we clearly observe experimentally and parametrise through Monte Carlo simulations. Our results provide a comprehensive understanding of the exciton physics in perovskite NCs and suggest possible optical control of the exciton states that could be employed for quantum information science.

## Results

### Zero-field exciton quantum beats

Due to the presence of heavy atoms, strong SOC in halide perovskites splits the energy bands with $J$-states, consisting of an upper $J_e = 3/2$ and lower $J_e = 1/2$ for the electron states in the conduction band while leaving the valance band almost unaffected ($J_h = 1/2$)[38]. Importantly, the total angular momentum $J$, as a consequence of SOC, is a mixture of up and down spins rather than pure spin states[16]. In the excitonic description, the total angular momentum $J = J_e + J_h$ of the exciton remains conserved while being threefold degenerate due to interaction with the center of mass (CoM) momentum of the exciton[25]. Taking asymmetry into account, the band edge $J = 1$ triplet exciton state then splits into sublevels with three orthogonal linear dipoles $\Pi_x$, $\Pi_y$, and $\Pi_z$,

where $\Pi_z$ is assumed to be optically inactive, being along the observing $z$-axis[25] as shown in Fig. 1a. In this study, we examine the cube shaped colloidal $CsPbBr_3$ NCs exhibiting an orthorhombic phase below room temperature (RT). As shown in Fig. 1b, these NCs have an average edge length of $5.5 \pm 0.8$ nm (hereafter termed NC5) with a size distribution which contributes to the broadening of absorption and PL linewidth (Fig. 1c). To monitor the exciton dynamics with spin decoherence present, CTA spectroscopy is utilized[39]. As a complementary measurement to the time-resolved Faraday/Kerr rotation, the time evolution of the net exciton spin polarization $S_z$, can be obtained by subtracting the kinetics obtained from two counter-circular probes following photoexcitation with the same circularly polarized pump (Supplementary Note 1).

Considering a simple two-level system without FSS, the left (right)-handed circularly polarized pump (denoted as $\sigma^+$ and $\sigma^-$, respectively) populates the lowest $J = 1$ state with excitons possessing angular momentum of $|+1\rangle$ ($|-1\rangle$). Upon a $\sigma^+$ ($\sigma^-$) probe, the blocked $|+1\rangle$ ($|-1\rangle$) transition due to phase-space filling and screening effects results in bleaching at the exciton resonance (X). In addition, the counter-circular configuration coherently generates $|+1\rangle|-1\rangle$ biexcitons, which are expected to be absent within the co-circular configuration because of the selection rule[40]. However, biexciton induced absorption is observed below bandgap with a red-shifted energy of ~2.39 eV for both circular probes as shown in Fig. 2a, b. The biexciton binding energy is then estimated to be ~50 meV at 12 K. At the same time, the photoinduced spectral shift and exciton linewidth change contribute to absorption above bandgap, for both co-circular and counter-circular geometries despite different intensities at the first few hundred femtoseconds (more details in Supplementary Fig. 9). The FSS has a strong influence on the exciton dynamics. Figure 2g schematically shows the simplified few-level quantum system and optical transitions in $CsPbBr_3$ NCs with the optical orientation of the excitons. The $\Pi_x$ and $\Pi_y$ linear dipoles, because of the superposition of the spin up and down excitons, are simultaneously excited by the phase-sensitive circular pump. Consequently, the cross-relaxation process is modulated periodically by the coherent superposition of $\Pi_x$ and $\Pi_y$ states, i.e., quantum beating between the two states. Figure 2c displays the quantum beating map which is obtained by subtracting the $\sigma^+$ and $\sigma^-$ probe following $\sigma^+$ pump, and picosecond optical oscillations are clearly resolved for the NC ensemble. Taking the exciton FSS into consideration (Supplementary Note 2), the calculated CTA contour plots shown in Fig. 2d–f are in full agreement with our experimental results.

Interestingly, quantum beats appear at both the exciton (X) and biexciton (XX) resonances, and at the above bandgap photoinduced absorption (PIA) peak as seen from the $S_z$ temporal profiles at two

**a**

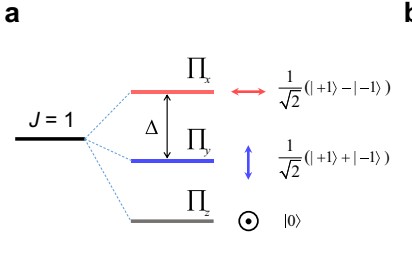

**b**

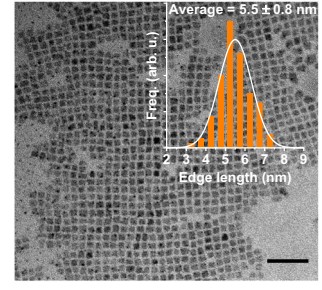

**c**

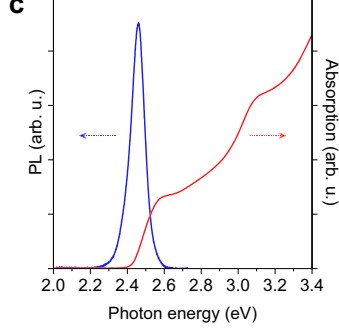

**Fig. 1 | Properties of CsPbBr₃ NCs. a** Fine structure splitting of the bright $J = 1$ exciton triplet state in orthorhombic $CsPbBr_3$ NCs. The degenerate $J = 1$ exciton state splits into three orthogonally and linearly polarized states $\Pi_x$, $\Pi_y$, and $\Pi_z$ as a result of the exchange interaction and shape anisotropy. The exciton spin up and down states are represented by $|+1\rangle$ and $|-1\rangle$, respectively. The laser beam direction and hence the spin polarization are along the $z$-axis. $\Delta$ denotes the energy splitting between the $\Pi_x$ and $\Pi_y$ states. **b** Transmission electron microscopy image of $CsPbBr_3$ NCs. Inset plot shows the size distribution. The black scale bar is 50 nm. **c** Room-temperature steady-state absorption and PL spectra of a NC5 ensemble.

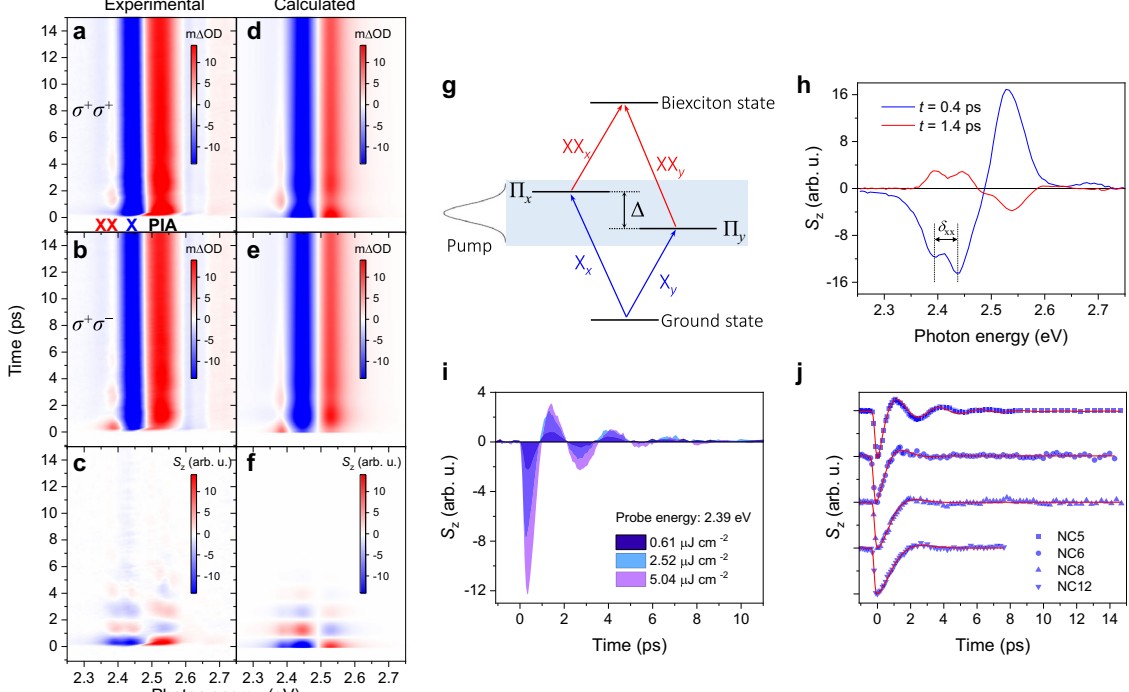

**Fig. 2 | Exciton quantum beating in CsPbBr₃ NCs upon resonant excitation at ~12 K. a** Contour plot of the time-resolved CTA spectra with co-circular configuration ($\sigma^+$ pump and $\sigma^+$ probe). XX, X, and PIA refer to the biexciton absorption, exciton bleaching, and photo-induced absorption, respectively. **b** CTA spectra with counter-circular configuration ($\sigma^+$ pump and $\sigma^-$ probe). **c** The exciton quantum beating map, which is the subtraction of $\sigma^+\sigma^+$ by $\sigma^+\sigma^-$ CTA spectra. **d**, **e**, and **f** the calculated CTA spectra corresponding to **a**, **b**, and **c**, respectively. **g** Schematic diagram of a few-level quantum system and optical transitions, where $X_{x/y}$ and $XX_{x/y}$ refer to exciton and biexciton occupation respectively. Here the pump pulse is broad enough to excite the two FSS states simultaneously. **h** Representative energy-resolved spectra at different delay times extracted from **c**, and $\delta_{XX}$ refers to the biexciton binding energy. **i** Exciton quantum beats with different pump excitation fluences, while probing at the biexciton resonance (~2.39 eV). **j** The net spin polarization dynamics for NCs with different sizes. The red solid lines represent the fits. Source data are provided as a Source Data file.

distinct beating times (Fig. 2h). We note that the phonon coherence cannot explain the π-phase shift between the two probes because the lattice vibration should not be sensitive to the light helicity in halide perovskites. From the beating map, an oscillation period of ~2.5 ps, corresponding to an energy of ~1.7 meV, can be estimated, which is ascribed to the energy difference between two linearly polarized eigenstates. Such an energy splitting is comparable to the reported values in the order of meV for CsPbBr₃ NCs[25,26] and CdSe QDs[41], while being larger than that for CsPbI₃ NCs (hundreds of μeV)[24,27,28] and arsenide-based QDs (tens of μeV)[7,42] measured by single-particle PL. In addition, the energy gap dependence of the splitting energy is also consistent with the literature reports[27,43], where enhanced splitting appears in the smaller NCs as shown by shorter oscillation periods in Fig. 2j. We note that our measurement provides the average splitting energy between any two linearly polarized eigenstates because of the random orientations (in such case the dipole orientation is immaterial in the substrate plane). Surprisingly, quantum beating can also be observed in NC ensemble by the linearly polarized pump-probe measurements where the subtracted signal exhibits a weaker oscillation with the same frequency to that from the circularly polarized measurements (Supplementary Fig. 12). Particularly, the linearly polarized measurement shows a longer decay due to the long-lived exciton population without the longitudinal spin relaxation process. Again, our horizontal pump interacts with both the $\Pi_x$ and $\Pi_y$ dipoles because of the random NC orientations. Hence the beating between the two coupled levels can be resolved. A single-particle level experiment may clarify the dipole alignments, which would allow the enhancing and erasing of the quantum beating according to the polarization distribution. Moreover, the independence of

oscillating frequency on the excitation density (Fig. 2i) indicates a robust coherence against many-body interactions.

## Exciton spin relaxation

The optical orientation of excitons allows the study of the spin lifetime $\tau_s$, i.e., the envelope of the oscillatory dynamics. To understand the exciton spin relaxation process, we examine the temperature dependences of $S_z$ in these CsPbBr₃ NCs. Figure 3a compares the spin-relaxation dynamics in an NC ensemble with an average particle size of $(6.3 \pm 1.3)$ nm (NC6) at different temperatures, where increasing temperature seems to prolong the spin-relaxation time up to RT and cancel out the beating signature. By fitting the net spin polarization dynamics with either a single-exponential (above 120 K) or a damped oscillation model (below 150 K), the exciton spin-relaxation time $\tau_s$ can be obtained, as summarized in Fig. 3b. For comparison, two other sizes $(8.3 \pm 1.4)$ nm (NC8), $(12.0 \pm 1.7)$ nm (NC12), and CsPbBr₃ polycrystalline thin film (PTF) samples are also inspected (see Supplementary Fig. 4–7 for sample characterizations). For the PTF sample, the spin dynamics versus temperature is interpreted in the context of the EY mechanism (Supplementary Note 3), where the increasing longitudinal optical (LO) phonon population with temperature leads to an acceleration of the spin relaxation. In contrast, the NC counterparts exhibit anomalous dependences below certain temperatures despite the shortened spin-relaxation times. As the temperature decreases from 340 K, the initial increase of $\tau_s$ can be attributed to the EY process. Although the EY lifetimes in NCs are shorter than in PTF, they display the same power-law fitting exponent of −1.3 with temperature. In this regime, decreasing NC size promotes faster spin relaxation, which is consistent with the literature[44]. All three NC

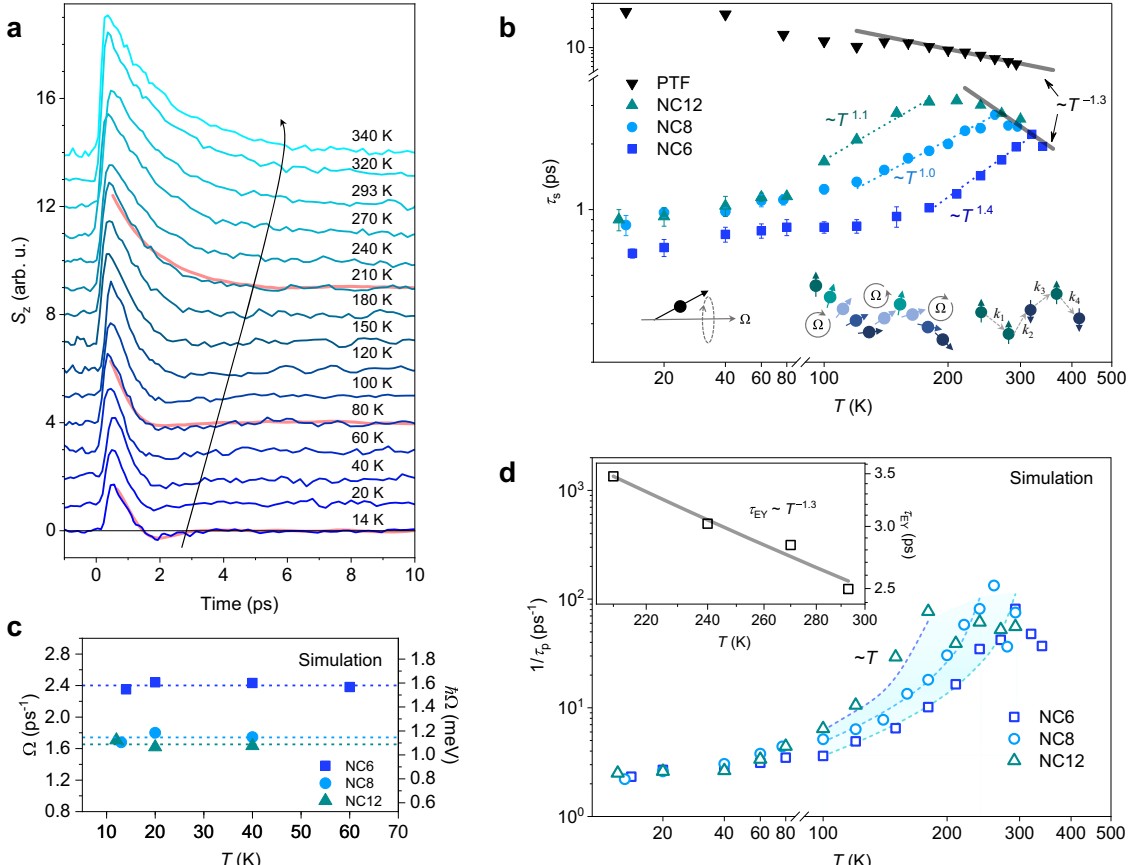

**Fig. 3 | Temperature dependences of the exciton spin lifetime in CsPbBr₃ NCs with resonant pumping and low excitation density. a** The net spin polarization dynamics at different temperatures for NC6 (for clarity, $S_z$ here is flipped to decay from a positive value). The simulated dynamics are selectively shown at 14 K, 80 K, and 210 K by red lines. **b** Extracted exciton spin lifetimes at different temperatures for NCs with three sizes and polycrystalline thin film. The error bars represent the standard errors of the fitted spin lifetimes. The gray lines with a power-law exponent of −1.3 are plotted for guidance. From left to right, the insets schematically show the spin precession along an effective magnetic field with a frequency of Ω,

the interruption of spin precession by changing the direction of the magnetic field via momentum scattering, and the EY process where the spin-flip may occur due to the change of momentum $k$. **c** The corresponding precession frequency and energy for different NCs extracted from the oscillatory dynamics at low temperatures. **d** Simulated momentum scattering rate for the motional narrowing process. Linear fits (dash lines and the shading) to the temperature (~T) are given for all three sizes above 100 K. The inset shows the simulated temperature dependence of the EY process. Source data are provided as a Source Data file.

ensembles show a $\tau_s$ maxima at around 200 K, 260 K, and 320 K for NC12, NC8, and NC6, respectively. The maxima indicate that an additional spin-relaxation channel is present in these NCs and manifests mainly at low temperatures, where it overtakes EY as the most efficient spin-relaxation mechanism.

The motional narrowing process is an alternative spin dissipation mechanism, where spin can coherently precess along an axis with its direction dependent on the quasiparticle's crystal momentum. The phenomenon of motional narrowing refers to the linewidth decrease of a resonant frequency due to the motion in an inhomogeneous system. Motional narrowing of spins has been demonstrated in nuclear magnetic resonance (NMR) and spin relaxation in III-V semiconductors. As a result, the motion narrows down the dephasing linewidth by interrupting the spin precession. There are two distinct characteristics of this precessional process. In the quasi-collision-free limit (*i.e.*, the scattering is significantly suppressed and ⟨Ω$τ_p$⟩>1, where Ω is the momentum-dependent Larmor precession frequency along the effective magnetic field and $τ_p$ the momentum scattering time) the spin undergoes full rotating cycles between scattering producing damped spin oscillation. This limit is usually found at low temperatures. On the other hand, in the strong scattering limit ⟨Ω$τ_p$⟩ ≤ 1 (*i.e.*, the spin cannot rotate full cycle between scattering events) the motional narrowing process leads to an exponential time decay of the spin, yet with the spin-relaxation time $τ_s$ having an inverse

relationship[45] with $τ_p$ that

$$\frac{1}{\tau_s} \propto \langle \Omega^2 \tau_p \rangle \qquad (1)$$

The latter effect, usually present at higher temperatures, is due to frequent interruptions of the spin precession between scattering events and often manifests in an anomalous temperature-dependence of the spin-relaxation time. The intermediate regime between these two limits shows a more complex relationship between $τ_p$ and $τ_s$.

By assuming $\Omega^2$ with a constant strength against temperature, we can deduce a $\tau_s \sim T^k$ ($k$>0) relationship with the condition that $\tau_p \sim T^{-k}$ (empirically, for LO phonon scattering) in the high-temperature range (above 100 K) as shown in Fig. 3b. Our results show that the exponents are close to these reported values for charge carriers in LHPs obtained by mobility measurements[46–49]. Since our results span both limits as well as the intermediate regime, simple fits of the experimental spin relaxation times cannot accurately describe the motional narrowing process. This is further aggravated by the contribution from the EY process.

To fully parametrise both the motional narrowing and the EY processes, we conducted Monte Carlo simulations (see Methods), which simultaneously account for both spin relaxation channels. At the highest temperatures the spin relaxation for all NC sizes is dominated

by the EY process. Interestingly all the NC sizes display the same EY lifetimes (above the temperatures where the spin dynamics is instead dominated by the motional narrowing process). In fact, we obtain the best fit assuming size-independent EY lifetimes with a $T^{-1.3}$ temperature dependence (see inset in Fig. 3d) across the full temperature range. As the temperature decreases, the dynamics crosses a regime where motional narrowing and EY scatterings contribute similarly to the spin decoherence, until a regime is reached where EY is too slow to significantly contribute to the dynamics.

The spin precession is assumed to be around in-plane momentum-dependent vectors of effective magnetic fields originating from the FSS energy splitting, which is changed in direction by the spin-preserving, momentum-randomizing scatterings. We assume a momentum-independent strength of the effective magnetic field with a constant spin precession frequency $\Omega$. Yet the momentum scattering rate $\tau_p^{-1}$ is assumed to be temperature and NC size dependent and individually fitted to each dataset. As shown in Fig. 3d, $\tau_p^{-1}$ is found to scale linearly with the temperature in the high-temperature region (Fig. 3d), verifying the LO phonon-dominated scattering mechanism as discussed before. Interestingly, $\tau_p^{-1}$ is independent of NC size and nearly constant at low temperatures, indicating scattering with acoustic phonons or/and defects. The precession frequency $\Omega$ can also be obtained from the low-temperature simulations as shown in Fig. 3c, and found, as expected, to be constant with temperature, consistent with the interpretation as quantum beating between the FSS levels. The average FSS energies of ~1.6 meV, ~1.2 meV, and ~1.1 meV for NC6, NC8, and NC12 can then be related by $\bar{\Delta} = \hbar\Omega$ respectively, in good agreement with Fig. 2j. The enhancement of the electron-hole exchange interaction or/and the Rashba effect in smaller NCs with high surface-to-volume ratio results in increased splitting energy and hence larger $\Omega$[27].

## Discussion

From high to low temperature, the exciton spin relaxation in CsPbBr$_3$ NCs can be divided into three regimes with different dominant mechanisms: 1) the EY mechanism where the scattering with LO phonons depolarizes spins; 2) the strong-scattered motional narrowing mechanism where the momentum scattering preserves spin polarization; and 3) the quasi-scattering-free motional narrowing of spins which yields oscillatory spin dynamics. These as-described exciton spin relaxation processes are schematically shown in the insets of Fig. 3b. Generally, in perovskite NCs, at high temperatures where the LO phonon scattering dominates, the EY process contributes as significantly to spin relaxation as it does in bulk materials. As the temperature decreases, the motional narrowing process becomes operative and competes with the EY process. Both DP and MSS mechanisms are of the motional-narrowing-type with different origins of the effective magnetic field, where the former is proposed for the conduction band electron spin relaxation in a SOC field in non-centrosymmetric semiconductors and their nanostructures[50], and the latter for the exchange-interaction-induced exciton spin relaxation[51] in 2D structures like monolayer MoSe$_2$[12] and quantum wells[36]. In perovskite NCs, an effective magnetic field originating from the FSS can be naturally established. Such intrinsic field must be momentum-dependent and in-plane as mentioned previously, which can be understood in terms of the Zeeman effect where the field strength can be deduced from the splitting energy. Consequently, CoM motion in momentum space randomizes the spin precession, which determines the relaxation process. Specifically, at rather high temperatures, the enhanced scattering yields varying directions of $\Omega$ and hence is expected to prolong the spin lifetime, which corresponds to the strong scattering regime in the motional narrowing process. However, in a quasi-collision-free regime (below ~100 K), the excitons are "localized" in momentum space and the spin

dynamics are oscillatory because of the quantum beating between two FSS levels with splitting energy of $\Delta$, at a frequency of $\Omega = \Delta/\hbar$. Such coherence between the FSS levels provides an upper limit for the spin dephasing rate.

In conclusion, our results provide a thorough understanding of the exciton coherence and spin relaxation in colloidal CsPbBr$_3$ nanocrystals, which will evoke further theoretical and experimental investigations of the exciton physics in perovskite nanostructures. In addition, we demonstrate that ultrafast spectroscopy allows estimating the exciton fine-structure splitting in ensembles of colloidal perovskite nanocrystals at zero magnetic field, even though that the exciton multi-levels are not spectrally resolved. However, such measurements can only provide average splitting because of the random dipole distribution and nanocrystal size distribution. Furthermore, the quantum beats may be smoothed with increasing temperature due to the motional narrowing process. On the other hand, single-particle PL measurements allow direct observation of the FSS levels with high energy resolution. Importantly, these FSS states are promising for quantum information science. For example, the two eigenstates $\Pi_x$ and $\Pi_y$, and the biexciton state can be engineered as a set of basis for logic devices[5]. The coexistence of the EY process and FSS-induced motional narrowing of exciton spin paves the way for spin engineering in perovskite nanostructures.

## Methods

### Synthesis of colloidal nanocrystals

Our CsPbBr$_3$ NC samples are approximately 5 nm (NC5), 6 nm (NC6), 8 nm (NC8) and 12 nm (NC12) in size. NC5 was synthesized by hot-injection according to a reported method with slight modifications[52]. Briefly, a Cs-Oleate stock solution was prepared by loading 0.815 g of Cs$_2$CO$_3$, 2.5 mL of oleic acid and 40 mL of octadecene into a round bottom flask and degassed under vacuum at 100 °C for 30 min. Thereafter, the solution was kept under nitrogen flow and the temperature was raised and kept at 120 °C prior to injection. In a separate round bottom flask, 75 mg of PbBr$_2$, 183.8 mg of ZnBr$_2$ along with 2 mL of oleic acid, 2 mL of oleylamine and 5 mL of octadecene were loaded and degassed under vacuum at 100 °C for 30 min. Then, the temperature was raised and kept at 170 °C until all salts dissolved. Subsequently, 0.5 mL of the prepared Cs-Oleate stock solution was swiftly injected and the reaction was quenched with an ice bath after 15 s. To separate the NCs, ethyl acetate was added to the crude solution in a volume ratio of 2:1 and the mixture became turbid. Then, the mixture was centrifuged at a Relative Centrifugal Force (RCF) of 11,627 × $g$ for 10 min. The supernatant was discarded, and the precipitate was redispersed in anhydrous toluene. Subsequently, the redispersed NCs were centrifuged at a RCF of 1860 × $g$ for 10 min and the precipitate was discarded. The supernatant containing NC5 was kept in the refrigerator for further use. Centrifuge rotor radius is 10.4 cm (Thermofisher Fiberlite™ F15-8x50cy).

The synthesis of NC6, NC8, and NC12 was performed according to a previously reported method under room temperature ambient conditions[53]. Briefly, a 0.1 M Cs-Octanoate stock solution was prepared by dissolving Cs$_2$CO$_3$ in octanoic acid. Separately, a PbBr$_2$ stock solution was prepared by dissolving 225 mg of PbBr$_2$, 2.9 g of trioctylphosphine oxide along with 0.2 M of octylphosphonic acid in 5 mL of anhydrous toluene. A 0.05 M solution of dimethyldidodecylammonium bromide (DDAB) was prepared by dissolving DDAB in anhydrous toluene. To prepare the NCs, 0.5 mL of the PbBr$_2$ stock solution was loaded into a vial under vigorous stirring. 55 μL of Cs-Octanoate stock solution was then swiftly injected. After the desired growth period (30 s, 2 min and 30 min for NC6, NC8 and NC12, respectively), 155 μL of DDAB solution was injected and the solution was left to stir for further 5 min. Thereafter, multiple centrifugations and redispersion steps were performed according to the published method. The resulting NCs were kept in the refrigerator for further use.

## Thin-film fabrication

For the polycrystalline thin-film, CsBr and $PbBr_2$ with a stoichiometric molar ratio of 1:1 are dissolved in dimethyl sulfoxide (DMSO) with a concentration of 0.2 M to form a precursor, which is then used for film deposition. Specifically, 20 µL of the precursor is dropped onto a 10 mm × 10 mm glass substrate for spin-coating with a speed of 4,000 r.p.m. followed by an annealing process at 100 °C for 10 min. All procedures are done in a nitrogen-filled glove box, and the as-deposited thin-film is of ~40 nm thick. For the NC solids, pre-synesized colloidal $CsPbBr_3$ NCs are directly used for spin-coating with a speed of 500–800 r.p.m., and the resultant thickness is tens of nanometers.

## Circularly-polarized transient absorption spectroscopy

The measurements were conducted using a commercial Helios™ transient absorption spectrometer from Ultrafast Systems LLC powered by an ~800 nm Libra™ Ti: Sapphire laser (~50 fs, 1 kHz) from Coherent Inc. Briefly, the output from the laser system is split into two beams with one directed into an optical parametric amplifier (Coherent OPeRa SOLO™) for energy-tunable pump and the other delayed one coupled onto a sapphire crystal for white light probe. The circularly polarized light was generated by achromatic quarter wave-plates and linear polarizers.

## Monte Carlo simulations

We simulate the net relaxation of the exciton spin polarization as the result of two competing scattering events which result in (i) motional narrowing and (ii) spin randomization, primarily attributable to the Elliott-Yafet (EY) mechanism[54]. Each exciton is represented by a 3-dimensional spin vector, which precesses about an effective in-plane magnetic field arising from the exciton-fine structure splitting, with a precession frequency $\Omega$ and a direction which lays on the sample's plane and depends on the center-of-mass momentum of the exciton. Apart from precession, the excitons can undergo two processes: a momentum randomizing motional narrowing scattering with a rate of $\tau_p^{-1}$ and a spin randomizing EY scattering with lifetime $\tau_{EY}$.

The spin dynamics are calculated using Monte Carlo over an ensemble of 2,000 excitons, and the parameters are extracted by best fitting with experimental results, performed using Powell minimization. While $\Omega$ and $\tau_p^{-1}$ are extracted independently for every sample and temperature, $\tau_{EY}$ are fitted to NC12 and then used for all samples and extrapolated at lower temperatures. Further details on the Monte Carlo simulations are collected in Supplementary Note 5 and 6.

## Data availability

Source data are provided with this paper. The data that support the findings of this study are also openly available in DR-NTU (Data) at https://doi.org/10.21979/N9/GEBAP3. Source data are provided with this paper.

## Code availability

The code that support the findings of this study are openly available in DR-NTU (Data) at https://doi.org/10.21979/N9/GEBAP3.

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

## Acknowledgements

This work is supported by the Ministry of Education under its AcRF Tier 2 grants (MOE2019-T2-1-006 (T.C.S.) and MOE-T2EP50120-0004 (T.C.S.)); and the National Research Foundation (NRF) Singapore under its NRF Investigatorship (NRF-NRFI-2018-04 (T.C.S.)) and the Competitive Research Programme (NRF-CRP25-2020-0004 (T.C.S.)). I.W. S.D.F. and M.B. acknowledge Nanyang Technological University, NAP-SUG (M.B.).

## Author contributions

T.C.S. and R.C. conceived the idea and designed the experiments. R.C. conducted the optical measurements with M.F., J.W.M.L, and D.G. R.C. calculated the optical spectra. J.W.M.L, R.C., and S.Y. prepared the samples. J.W.M.L performed the transmission electron microscopy characterization. M.B., I.W., and S.D.F. conducted the Monte Carlo simulations. R.C., T.C.S., M.B., and I.W. analysed the data. R.C. drafted the manuscript and T.C.S. and M.B. revised it. All authors discussed the results and commented on the manuscript at all stages. This project is led by T.C.S.

## Competing interests

The authors declare no competing interests.
