## [Peer Review File · Nature Communications]

REVIEWER COMMENTS

Reviewer #1 (Remarks to the Author):

In this manuscript, the authors reported the phenomenon of quantum beating between two exciton splitting states in the perovskite nanocrystals and discussed the spin polarization relaxation mechanism in different temperature regimes. The authors have done a thorough study on the spin depolarization, including temperature dependent and crystal size dependent measurements, and carefully examined the underlying mechanisms. The results are well presented, and the conclusion is sound. However, the following concerns and comments should be addressed prior to considering for publication.

1. The splitting of the degenerate exciton states was attributed to the shape anisotropy of the nanocrystal. First, if this is true, a larger size distribution of the nanocrystals (i.e., likely give rise to a smaller shape anisotropy) should result in a relatively broad distribution of the corresponding energy levels, and given such a small splitting, the oscillation of the TA kinetics arising from the quantum beating will likely be smoothed. Thus, the reviewer is wondering if the size distribution of the nanocrystals affects the amplitude of the oscillation amplitude. Second, the statistics of the dimensions of the nanocrystal should be provided, which can serve as evidence of the shape anisotropy.
2. "Motional narrowing process" should be further elaborated, and it is difficult for readers to comprehend this concept without any background introduction in the context.
3. The authors claim that the spin precess about the effective magnetic field. The exciton state splitting was employed to estimate the strength of this field, and the reviewer is wondering what is the nature of this effective magnetic field and why this field is related to the momentum of the exciton. It would be the best that further discussion about this effective magnetic field is provided.
4. The effective magnetic field (line 235) was accidentally denoted as Ω . It is quite confusing because the Larmor precession frequency was denoted as Ω , too.
5. In Fig. 2h, why the splitting was only shown on the higher energy side of the derivative-like feature.

Reviewer #3 (Remarks to the Author):

Present manuscript reports on experimental studies of lead-halide perovskite, CsPbBr₃, nanocrystals (NCs). The focus is exciton coherent dynamics within the band-edge J=1 exciton triplet whose fine (FSS) structure splitting is attributed to the Rashba effect. The authors use circularly polarized transient absorption (CTA) spectroscopy to probe quantum beats between selected two orthogonally polarized states associated with symmetric and anti-symmetric linear combinations of $J_z = -1, 1$ exciton states. The authors claim that this is the first reported use of a coherent spectroscopy to directly resolve quantum beats (~ 4 ps) within the exciton FSS (splitting of 1.6 meV). Subsequent temperature resolved examination of the quantum beat decay time attributed to the spin relaxation time shows a crossover of different spin relaxation dynamics. The authors interpret them to the Elliott-Yafet (EY) at high-T and motional narrowing at low-T. A Mont Carlo simulation results are reported in support of such an interpretation.

Direct observation of the quantum beats within the optically unresolved FSS and experimental identification of different spin-relaxation dynamics as a function of temperature represent strong points of this study. However, the manuscript has several drawbacks that the authors need to address.

Comment 1. The manuscript presentation needs significant improvements. Some statements lack clarity, some are contradictory:

L.110: Reported ~ 2.39 eV is red shifted bi-exciton energy not the bi-exciton energy detuning as stated. The detuning in this case should be the exciton binding energy not even specified in the manuscript. It'll be useful to know values for the binding energy and make the statement more accurate.

LL. 111-113: The statement that "the photoinduced spectral shift and exciton linewidth changes contribute to the absorption above bandgap" does not explain observed above bandgap absorption but rather raises another equation how those changes contribute to the absorption effect. The authors need to clarify this.

LL. 126-127: Like above, why the pi-phase shift eliminates the phonon contribution to the exciton coherences?

LL.134-137. "We note that our measurement provides the average splitting energy between two linearly polarized eigenstates because of the random orientation of exciton's CoM momentum (in such case P_{ix} and P_{iy} dipole orientation is immaterial)". This statement is rather confusing. P_{ix} and P_{iy} dipole orientations should be determined by the crystallographic symmetries, the randomization

of the CoM momentum averages during the measurements and the response should be consistent with P_{ix} and P_{iy} momentum orientation. The authors need a better explanation of what the “average splitting” means.

In LL. 215-216, the authors assume time independent Larmor frequency Ω to extract the power-law exponents for momentum relaxation. Below in LL.235-236 they claim that Ω was assumed to be temperature dependant in the MC calculations. These statements contradict to each other.

In L. 258: the intrinsic magnetic field in perovskite NCs “must be momentum dependant”, however, LL. 234-234 make a contradictory statement that in the MC simulations “We assume momentum-independent strength of the effective magnetic field Ω .” This needs to be clarified.

LL.248-249 state that the exciton spin relaxation dynamics can be divided into three regimes. From the discussion below it is hard to say what those three regimes are. It'll be helpful for the reader if these three regimes are clearly itemized (in addition see related Comment 4).

LL 279-280. “The large bi-exciton binding energy help the coherence survive at high temperature”. This concluding statement seems totally misaligned with the manuscript main theme focused on the spin relaxation processes due to the exciton center of mass motion. The electron-hole exchange processes are contributing to both the exciton/bi-exciton binding energies and to the spin relaxation dynamics. However, the manuscript has no clear study of these contributions not even a discussion about the relationship between the X/XX binding energies and relaxation mechanisms.

Supplementary Information Eq. S4 contains a sum of different contributions to the CTA signal, however, only bi-exciton absorption term is given in Eq. S3. The authors should list the rest of the terms.

Comment 2: The interpretation of the quantum beat decay is solely given based on the spin-flip, i.e., population decay T1-dynamics. This justifies the rate equations S1 and S2 used by the authors. However, pure-dephasing T2 mechanisms can be involved into the quantum beats decay which is common to consider in the pump-probe spectroscopies. In the NMR and indeed in visible/optical range such an interplay is accounted for by describing the dynamics with the help of the Redfield equation. The authors need to provide solid arguments towards eliminating the pure dephasing mechanisms of the quantum beat decay from consideration.

Comment 3: Bringing out linearly polarized line measurements in LL. 137-145 raises questions of why the quantum beat decay shows down, what relaxation mechanisms are involved and how this is aligned with the CTA signal interpretation. The questions are left open and never revisited in the manuscript.

Comment 4: In the discussion of possible motional narrowing mechanisms, the authors refer to the D'yakonov-Perel' (DP) and Maijalla-Silva-Sham (MSS) mechanisms. The latter seems to be important and is brought into consideration in analogy to detailed study of 2D lead-halide perovskites nanoplates (NPs) Ref. 35. That study concludes that the dynamic polaritonic screening facilitates the MSS mechanism in the NPs. In contrast, in the NCs the exciton-polariton effects are suppressed (see, e.g., Section S1G in Ref. 23). It is reasonable to expect that the Rashba effect could be a contributor to the MSS. This aspect has not been studied in the NCs and, therefore, the manuscript lacks detailed examination of the MSS mechanism and comparison with the results for 2D NPs, Ref. 23. If the authors provide such an examination, the impact of the paper will be enhanced to the level required by the Nature Communication.

The work studies ensembles of CsPbBr₃ nano crystals with circularly-polarized transient absorption (CTA), and reports zero-field exciton quantum beating. The exciton fine structures in these lead halide perovskites nano crystals has been under debated, especially the possibility of large Rashba effect. This study should be of great interest, and its observations are also new – oscillations of S_z at zero magnetic field in CTA measurement. However, the interpretation is deficient, and data is not sufficient to draw the conclusions.

1. The oscillations in Fig. 2c are interpreted as quantum beating between “two linearly polarized eigenstates”, or “coherent superposition of Π_x and Π_y states. But the authors did not give any clear explanation how the CTA measurement could measure splittings between the two states, in terms of optical transition strengths and symmetries of these states. For example, in the Supplementary Note 1, Equation (S1) and (S2) directly start with population for $|+1\rangle$ and $|-1\rangle$ states, not the eigenstates Π_x , Π_y , and Π_z . The authors should carefully work out the Hamiltonian, taking into parameters for spin exchange interaction and anisotropy due to crystal field, and find the signal for two or three major crystal orientations. Probably also discuss effect of averaging different crystal orientations.

If my understanding is right, CTA is a circular dichroism measurement with circularly polarized pump. In zero field, the (frequency of) oscillations likely measures energy splitting corresponding to exchange couplings, rather than splitting due to anisotropy in crystal field. Furthermore, for the picture of Fig. 1a, a more suitable measurement should be probed by linear dichroism with a linear pump. The author probably showed some data in Fig. S12, mentioning weaker oscillations with same frequency, but there is no further systematic measurements. For both circular dichroism and linear dichroism measurements, it's worth applying magnetic field and directly verify the contribution of Zeeman terms.

Some more details for the authors to check: in the main text, the oscillation period is said to be ~ 4.1 ps, but the period in Fig. 2c is only about 2 ps; it's not clear how to get the the corresponding energy of ~ 1.6 meV either.

2. The main conclusions of this manuscript include the anomalous temperature dependence of the spin lifetime and explanation of spin relaxation mechanisms in different temperature ranges. The authors have to justify the time scales extracted from fitting the time traces be spin relaxation times, and how the measured dynamics is not dominated by inhomogeneous broadening on ensemble of NCs.

Related to both of the points, there is one useful reference, Yugova *et al*, “Exciton fine structure in InGaAs/GaAs quantum dots revisited by pump-probe Faraday rotation” PRB 75, 195325 (2007).

Response to the Reviewers

We would like to thank all the Reviewers for their comments, which has helped to improve our manuscript. We have made our best efforts, to address all the Reviewers' concerns and revise the manuscript accordingly. In this letter, comments from reviewers are given in *green italics*, our responses are given in black, our highlighted revisions are given in *blue italics*, and deleted sections from the original manuscript are given in *red italics*.

Reviewer #1

In this manuscript, the authors reported the phenomenon of quantum beating between two exciton splitting states in the perovskite nanocrystals and discussed the spin polarization relaxation mechanism in different temperature regimes. The authors have done a thorough study on the spin depolarization, including temperature dependent and crystal size dependent measurements, and carefully examined the underlying mechanisms. The results are well presented, and the conclusion is sound. However, the following concerns and comments should be addressed prior to considering for publication.

We thank the Reviewer for the encouraging comments. Our response to the detailed comments is as follows:

- 1. The splitting of the degenerate exciton states was attributed to the shape anisotropy of the nanocrystal. First, if this is true, a larger size distribution of the nanocrystals (i.e., likely give rise to a smaller shape anisotropy) should result in a relatively broad distribution of the corresponding energy levels, and given such a small splitting, the oscillation of the TA kinetics arising from the quantum beating will likely be smoothed. Thus, the reviewer is wondering if the size distribution of the nanocrystals affects the amplitude of the oscillation amplitude. Second, the statistics of the dimensions of the nanocrystal should be provided, which can serve as evidence of the shape anisotropy.*

We thank the Reviewer for raising this important question. Indeed, the size distribution of NCs affects the oscillating amplitude. As shown in Eq. R1 (as well as Eq. R24~25 in this letter), the oscillating amplitude at the exciton/biexciton resonance (also the defined net spin polarization S_z) is inversely proportional to the FSS dipole transition linewidth $\Gamma_{deph.}$:

$$S_z \propto -\frac{4}{\Gamma_{deph.}} e^{-\frac{t_D}{\tau_{xy}}} \cos \omega_{xy} t_D \quad (\text{R1})$$

with the quantum beat decay time τ_{xy} and frequency ω_{xy} (we have used the symbol Ω in the Main text). Therefore, the inhomogeneous broadening of the dipole transition linewidth due to the size distribution decreases the oscillating amplitude. Moreover, the beating behaviour will be smoothed out if the size distribution is too large (which means that the averaging of the

beating frequency must be considered). To further support our statements, we have intentionally mixed two sets of NCs with slightly different average sizes (Fig. R1a). As shown in Fig. R1b, the main bleaching peak for both NC_{5.1} and NC_{5.5} is much more pronounced with less broadening, as compared to the mixture which has a bleaching peak position in the middle of these individual sets of NCs and possess a much larger degree of broadening stemming from the size distribution. Clearly as seen in Fig. R1c, the beating amplitude decreases for the mixture, which is consistent with our interpretation. Therefore, our conclusions are not affected by the small size distribution of our samples.

Fig. R1: CTA spectra and dynamics of NCs with different average sizes and their mixture (counter-circular geometry) at 12 K. The excitation energy is fixed at ~ 2.47 eV for all three samples. **a** Size distribution for the two sets of NCs used for mixture. **b** The CTA spectra for NC_{5.5} which is the same one used for Fig. 2 in the Main text (NC5), NC_{5.1} with a slightly smaller average size than NC_{5.5}, and the mixture of NC_{5.1} and NC_{5.5}. **c** Decay dynamics around the biexciton resonances.

We have also indicated the statistics of the dimensions of NCs in the TEM images (the insets in Fig. 1b and Fig. S6). Specifically, (6.3 ± 1.3) nm for NC6, (8.3 ± 1.4) nm for NC8, (12.0 ± 1.7) nm for NC12 respectively, which have been added in the revised manuscript.

2. *“Motional narrowing process” should be further elaborated, and it is difficult for readers to comprehend this concept without any background introduction in the context.*

We thank the Reviewer for raising this comment. We have added further elaboration of the motional narrowing process in the revised manuscript:

Page 7, Line 190-194:

The phenomenon of motional narrowing refers to the linewidth decrease of a resonant frequency due to the motion in an inhomogeneous system. Motional narrowing of spins has been demonstrated in nuclear magnetic resonance (NMR) and spin relaxation in III-V semiconductors. As a result, the motion narrows down the dephasing linewidth by interrupting the spin precession.

3. *The authors claim that the spin precess about the effective magnetic field. The exciton state splitting was employed to estimate the strength of this field, and the reviewer is wondering what is the nature of this effective magnetic field and why this field is related to the momentum of the exciton. It would be the best that further discussion about this effective magnetic field is provided.*

We thank the Reviewer for his comment on the origin of the effective magnetic field. For a single NC with symmetry breaking along the z-axis, two FSS states Π_x and Π_y can be understood in terms of the Zeeman effect due to an effective magnetic field in the x-y plane. The splitting energy between them is then used to estimate the field strength. We have used Eq. S25 for fitting to extract the beating frequency and hence the splitting energy. The effective magnetic field strength is related to the momentum due to the band dispersion. We have added these discussions about the effective magnetic field accordingly in the revised manuscript:

Page 10, Line 270-275:

In perovskite NCs, an effective magnetic field originating from the FSS can be naturally established. Such intrinsic field must be momentum-dependent and in-plane as mentioned previously, which can be understood in terms of the Zeeman effect where the field strength can be estimated from the splitting energy. Consequently, CoM motion in momentum space randomizes the spin precession, which determines the relaxation process.

4. *The effective magnetic field (line 235) was accidentally denoted as Ω . It is quite confusing because the Larmor precession frequency was denoted as Ω , too.*

We thank the Reviewer for pointing out this issue. To avoid this confusion, we have amended this part in the revised manuscript:

Page 9, Line 241-242:

“We assume a momentum-independent strength of the effective magnetic field Ω ” has been changed to “We assume a momentum-independent strength of the effective magnetic field with a spin precession frequency Ω ”.

5. *In Fig. 2h, why the splitting was only shown on the higher energy side of the derivative-like feature.*

We think that there might be some misunderstanding here. The source of confusion could be due to Fig. 2g where we use Δ to denote the fine structure splitting energy between Π_x and Π_y states, which is very small (\sim meV) and is not spectrally resolved here. In Fig. 2h, we show the spectra at two different time delays (0.4 ps and 1.4 ps) to highlight the beating signature. The “splitting” features the exciton transition (\sim 2.46 eV) and biexciton transition (\sim 2.39 eV), and the biexciton binding energy was denoted as Δ_{xx} . The high-energy side peak originates from

photo-induced absorption. In light of this potential confusion, we have changed Δ_{xx} to E_{XX} referring to the biexciton binding energy in Fig. 2h:

Page 6, Line 152:

Reviewer #2

The work studies ensembles of CsPbBr₃ nano crystals with circularly polarized transient absorption (CTA), and reports zero-field exciton quantum beating. The exciton fine structures in these lead halide perovskites nano crystals have been under debated, especially the possibility of large Rashba effect. This study should be of great interest, and its observations are also new - oscillations of S_z at zero magnetic field in CTA measurement. However, the interpretation is deficient, and data is not sufficient to draw the conclusions.

We thank the Reviewer for the comments. As pointed out by the Reviewer, the effect of the Rashba splitting in perovskite NCs is still under debate. However, a study on how the Rashba effect affects the FSS is beyond the scope of our work. We have provided more data to support our conclusions on the zero-field quantum beats and spin decoherence mechanisms.

1. The oscillations in Fig. 2c are interpreted as quantum beating between “two linearly polarized eigenstates”, or “coherent superposition of Π_x and Π_y states”. But the authors did not give any clear explanation how the CTA measurement could measure splittings between the two states, in terms of optical transition strengths and symmetries of these states. For example, in the Supplementary Note 1, Equation (S1) and (S2) directly start with population for $|+1\rangle$ and $|-1\rangle$ states, not the eigenstates Π_x , Π_y , and Π_z . The authors should carefully work out the Hamiltonian, taking into parameters for spin exchange interaction and anisotropy due to crystal field, and find the signal for two or three major crystal orientations. Probably also discuss effect of averaging different crystal orientations.

We thank the Reviewer for this feedback. From a technical approach, we define the spin-relaxation time in Supplementary Note. 1 where the $|\pm 1\rangle$ exciton spins are optically injected. This is reasonable for estimating the spin relaxation with exponential decay. In Supplementary Note 2, we take the FSS into consideration and derive the modulated spin dynamics (*i.e.*, the role of the eigenstates Π_x , Π_y , and Π_z). The low-temperature spin lifetimes can then be extracted. *We also wish to highlight that our ensembles of NCs are different from conventional self-assembled quantum dots (QDs) that are grown by molecular beam epitaxial (MBE) with deterministic growth direction.* In our case, the perovskite NCs are spin-coated onto substrates with random orientations. Nonetheless, we have noticed the lack of specificity on signal detection in Supplementary Note. 2. Therefore, to make the discussion clearer, we start by treating single NCs with determined crystal orientations as suggested. Detailed theoretical analysis of the bright exciton splitting for perovskite NCs has already been reported in recent years (*Phys. Rev. B*, 97, 235304 (2018); *Nature* 553, 189-193 (2018)). Based on recently reported single-particle measurements (*Phys. Rev. Lett.*, 119, 026401(2017); *Nature* 553, 189-193 (2018); *Nat. Mater.* 18, 717-724 (2019); *Science*, 363 (6431), 1068-1072 (2019)), and mutually orthogonal Π_x , Π_y , and Π_z dipoles can be used in our framework. For generalization, we consider an orthorhombic distortion of D_{2h} symmetry which splits the $J = 1$ exciton states into three non-degenerate states, that are linearly polarized along the three symmetry axes,

respectively. As shown in Fig. R2, coherently excited superposition of any in-plane dipoles (here in-plane means the substrate plane that is perpendicular to the laser beam direction) allows us to observe the quantum beats (either by linearly polarized pump that is not perfectly aligned to one dipole, or by circularly polarized pump that inject spins at the same time). This explains the observation of beating under linearly polarized measurements. Regardless of the size distribution and tilted orientations, the manifested quantum beats measure the average splitting energies between any two FSS levels due to the random orientations. Next, we calculate the beats-modulated TA spectra with the condition of Fig. R2a.

Fig. R2: Nanocrystals with different crystal orientations on substrate

In light of the reviewer's feedback, the following treatment has also been added to Supplementary Note. 2 in the revised Supported Information.

The dynamics of the quantum system is initially calculated with an external potential from the in-coming light fields via a Liouville equation. Then the probe field change (and hence the TA signal) that is induced by the polarization field originating from the dynamics of the excited states, is calculated by the macroscopic Maxwell equations.

The optically induced dynamics of the system with optical stimulus and the relaxation processes can be described by the quantum mechanical Liouville equation with Lindblad dissipator, and the density matrix of the interaction between two states a and b can be approximated as

$$\dot{\rho}_{a,b}(t) = -i \frac{1}{\hbar} ([\hat{H}^0 + \hat{H}^\gamma(t), \hat{\rho}(t)])_{a,b} + D_{a,b}(\hat{\rho}) \quad (R2)$$

where \hat{H}^0 and $\hat{H}^\gamma(t)$ refer to the stationary energies of levels and the influence of the optical fields, and $D_{a,b}(\hat{\rho})$ the Lindblad dissipator. The dynamics starts with full occupation of the exciton ground state and are separated temporally.

1) Dynamics during the laser pulses

The light-matter interaction can be described with the typical dipole transitions and rotating wave approximation (RWA) that

$$\hat{H}^\gamma(t) = -\mathbf{E}_{in}^+(t) \cdot \hat{\mathbf{d}}^+ - \mathbf{E}_{in}^-(t) \cdot \hat{\mathbf{d}} \quad (R3)$$

with the dipole operators $\hat{\mathbf{d}}^+$ and $\hat{\mathbf{d}}$, describing the excitation and relaxation respectively, and the electric field $\mathbf{E}_{in}^\pm = \mathbf{E}_{pump}^\pm + \mathbf{E}_{probe}^\pm$, where

$$\mathbf{E}_{pu/pr}^+(t) = (k_x^{pu/pr} \mathbf{e}_x + k_y^{pu/pr} \mathbf{e}_y) E_{pu/pr}^0(t) e^{-i(\omega_{pu/pr} t + \varphi_{pu/pr})} \quad (R4)$$

$$\mathbf{E}_{pu/pr}^-(t) = (k_x^{pu/pr} \mathbf{e}_x + k_y^{pu/pr} \mathbf{e}_y) E_{pu/pr}^{0*}(t) e^{i(\omega_{pu/pr} t + \varphi_{pu/pr})} \quad (R5)$$

with defined light polarization $k_x^{pu/pr} \mathbf{e}_x + k_y^{pu/pr} \mathbf{e}_y$ ($k_x^{pu/pr} = 0$ for horizontal polarization and $k_y^{pu/pr} = 0$ for vertical polarization, and $k_{pu/pr}(\mathbf{e}_x \pm i\mathbf{e}_y)$ for circular polarization).

The relaxation and dephasing are neglected to calculate the dynamics during short laser pulses,

within the interaction picture $\tilde{\rho}_{a,b}(t) = e^{\frac{i}{\hbar}(H_{a,a}^0 - H_{b,b}^0)t} \rho_{a,b}(t)$ and then the remaining equation of motion

$$\dot{\tilde{\rho}}_{a,b}(t) = -i \frac{1}{\hbar} \sum_j \left(\tilde{H}_{a,j}^{\gamma, pu/pr}(t) \tilde{\rho}_{j,b}(t) - \tilde{\rho}_{a,j}(t) \tilde{H}_{j,b}^{\gamma, pu/pr}(t) \right) \quad (R6)$$

where

$$\tilde{H}_{j,b}^{\gamma, pu/pr}(t) = H_{j,b}^{\gamma, pu/pr}(t) e^{i\omega_{j,b} t} = -E_{pu/pr}^0(t) e^{-i(\varphi_{pu/pr} + \omega_{j,b} t)} \tilde{\mathbf{J}} \cdot \tilde{\mathbf{d}}_{j,b}^+ + h.c. \quad (R7)$$

with *h.c.* the Hermitian adjoint term. With an approximation of the jump condition, the effect on the density matrix by the pulse can be described as

$$\hat{\rho}^{\text{after } pu/pr}(t) = e^{-\frac{i}{\hbar} \hat{\Lambda}^{\gamma, pu/pr}} \hat{\rho}^{\text{before } pu/pr} \left(e^{-\frac{i}{\hbar} \hat{\Lambda}^{\gamma, pu/pr}} \right)^+ \quad (R8)$$

in which

$$\Lambda_{j,b}^{\gamma, pu/pr} = -\int E_{pu/pr}^0(t) dt \cdot e^{-i\varphi_{pu/pr}} (k_x^{pu/pr} \mathbf{e}_x + k_y^{pu/pr} \mathbf{e}_y) \cdot \mathbf{d}^+ \tilde{\delta}_{\omega_{pu/pr}, \omega_{j,b}} + h.c. \quad (R9)$$

with $\tilde{\delta}_{\omega_{pump}, \omega_{x/y, GS}} = 1$ stands for the transition between the ground state and the *x/y* state upon pump pulse and $\tilde{\delta}_{\omega_{probe}, \omega_{XX, x/y}} = 1$ between the *x/y* state and the biexciton state upon probe pulse respectively, and $\tilde{\delta}_{\omega_{pu/pr}, \omega_{j,b}} = 0$ for other *j* values. For perfectly linearly polarized transitions $\langle a | \mathbf{d} | b \rangle = d_0 \mathbf{e}_{x/y}$, the non-vanishing matrix elements of $\hat{\Lambda}^{\gamma, pu/pr}$ writes

$$\hat{\Lambda}^{\gamma, pu/pr} = -\int E_{pu/pr}^0(t) dt \cdot \begin{pmatrix} GS & y & x & XX \\ 0 & e^{i\varphi_{pu/pr}\mu_y^*} & e^{i\varphi_{pu/pr}\mu_x^*} & 0 \\ e^{-i\varphi_{pu/pr}\mu_y} & 0 & 0 & e^{i\varphi_{pu/pr}\nu_y^*} \\ e^{-i\varphi_{pu/pr}\mu_x} & 0 & 0 & e^{i\varphi_{pu/pr}\nu_x^*} \\ 0 & e^{-i\varphi_{pu/pr}\nu_y} & e^{-i\varphi_{pu/pr}\nu_x} & 0 \end{pmatrix} \quad (R10)$$

with $\mu_{x/y} = k_{pu}^{x/y} d_0$ and $\nu_{x/y} = k_{pr}^{x/y} d_0$ (For simplicity, we use d_0 for both x and y states).

2) Dynamics between and after the pulses

The contributions from the Lindblad dissipator must be considered to calculate the dynamics between the pulses, which can be separated into the relaxation processes and pure dephasing

$$\hat{D}(\hat{\rho}) = \hat{D}^{relax.}(\hat{\rho}) + \hat{D}^{deph.}(\hat{\rho}) \quad (R11)$$

Since the exciton population time is much longer than the oscillating period, the relaxation between x and y states can be neglected considering the time averaging. Therefore, the population relaxation processes can be described by $D_{x/y,x/y}^{relax.} = -\gamma_{x/y,x/y} \rho_{x/y,x/y}$,

$D_{GS}^{relax.} = \gamma_{x,x} \rho_{x,x} + \gamma_{y,y} \rho_{y,y}$, with defined $\gamma_{x,x} = \gamma_{y,y} = \frac{1}{\tau_x} = \frac{1}{\tau_y} = \frac{1}{\tau_r}$ (where $\tau_x = \tau_y = \tau_r$ means

the equal exciton population lifetime). The dephasing terms consist of the quantum beats dephasing $D_{x,y}^{deph.} = -\frac{1}{\tau_{xy}} \rho_{x,y}$, and the interband pure dephasing $D_{GS,x/y}^{deph.} = -\frac{1}{\tau_{\delta_{GS,x/y}}} \rho_{GS,x/y}$ and

$D_{x/y,XX}^{deph.} = -\frac{1}{\tau_{\delta_{x/y,XX}}} \rho_{x/y,XX}$ where τ_{δ} refers to the pure dephasing time.

Between the pump and probe pulses, the term $\hat{H}^{\gamma}(t)$ in Eq. R2 vanishes and only the free evolution due to \hat{H}^0 and $D_{a,b}(\hat{\rho})$ are considered. For the remaining dephasing and relaxation processes, an analytical solution can be given in the block-diagonal form

$$\hat{\rho}(t) = \begin{pmatrix} 0 & 0 \\ 0 & \hat{\rho}_{measure}(t) \end{pmatrix} \quad (R12)$$

where

$$\hat{\rho}_{\text{measure}}(t) = \begin{pmatrix} \text{GS} & y & x & \text{XX} \\ 1 - \rho_{y,y}(t_0)e^{-\frac{t}{\tau_r}} - \rho_{x,x}(t_0)e^{-\frac{t}{\tau_r}} & \rho_{\text{GS},y}(t_0)e^{i\omega_y \text{GS}t} e^{-\frac{t}{\tau_{\text{GS}}}} & \rho_{\text{GS},x}(t_0)e^{i\omega_x \text{GS}t} e^{-\frac{t}{\tau_{\text{GS}}}} & 0 \\ \rho_{y,\text{GS}}(t_0)e^{-i\omega_y \text{GS}t} e^{-\frac{t}{\tau_{\text{GS}}}} & \rho_{y,y}(t_0)e^{-\frac{t}{\tau_r}} & \rho_{y,x}(t_0)e^{i\omega_y t} e^{-\frac{t}{\tau_y}} & \rho_{y,\text{XX}}(t_0)e^{i\omega_y \text{XX}t} e^{-\frac{t}{\tau_{\text{XX}}}} \\ \rho_{x,\text{GS}}(t_0)e^{-i\omega_x \text{GS}t} e^{-\frac{t}{\tau_{\text{GS}}}} & \rho_{x,y}(t_0)e^{-i\omega_y t} e^{-\frac{t}{\tau_y}} & \rho_{x,x}(t_0)e^{-\frac{t}{\tau_r}} & \rho_{x,\text{XX}}(t_0)e^{i\omega_x \text{XX}t} e^{-\frac{t}{\tau_{\text{XX}}}} \\ 0 & \rho_{\text{XX},y}(t_0)e^{-i\omega_y \text{XX}t} e^{-\frac{t}{\tau_{\text{XX}}}} & \rho_{\text{XX},x}(t_0)e^{-i\omega_x \text{XX}t} e^{-\frac{t}{\tau_{\text{XX}}}} & \rho_{\text{XX},\text{XX}}(t_0) \end{pmatrix} \quad (\text{R13})$$

with the respective previous laser pulse time t_0 .

The induced polarization can then be calculated via $\mathbf{P}(t) = \text{Tr}(\hat{\mathbf{d}}\hat{\rho}(t))$, with which the signal field out from the NC $\mathbf{E}_{\text{out}}(t)$ can be calculated by the macroscopic Maxwell equations that

$$\mathbf{E}_{\text{out}}(t) = \mathbf{E}_{\text{in}}(t) + i \frac{\omega_c \mu_0 c l}{2} \mathbf{P}(t) \quad (\text{R14})$$

where ω_c, μ_0, c, l denote the laser frequency, vacuum permeability, the speed of light, and the NC edge length along z-axis.

The intensity of the signal field in frequency domain is then

$$I_{\text{out}}(\omega) \sim |\mathbf{E}_{\text{out}}(\omega)|^2 = |\mathbf{E}_{\text{in}}(\omega)|^2 + \omega_c \mu_0 c l \text{Im}(\mathbf{E}_{\text{in}}(\omega) \mathbf{P}^*(\omega)) + \frac{1}{4} |\omega_c \mu_0 c l \mathbf{P}(\omega)|^2 \quad (\text{R15})$$

The heterodyne detection allows a good approximation of the TA signal that

$$\Delta A(\omega) \sim -\omega_c l \text{Im}[\mathbf{E}_{\text{in}}(\omega) \mathbf{P}^*(\omega)] \quad (\text{R16})$$

The signal around the biexciton resonances (XX) can be calculated by

$$\Delta A_{x/y}^{\text{XX}}(\omega) \sim \text{Im}[\mathbf{E}_{\text{in}}(\omega) \mathbf{P}_{x/y}^{*\text{XX}}(\omega)] \quad (\text{R17})$$

where $\mathbf{P}_{x/y}^{*\text{XX}}(\omega) = \mathcal{F}(\rho_{\text{XX},x/y}(t) d_0 \mathbf{e}_{x/y})(\omega)$ by Fourier transform.

For a given time delay t_D between the pump and probe pulses, the solution gives the TA signal for the y dipole that

$$\begin{aligned}
\Delta A_y^{XX}(\omega, t_D) \sim & -\frac{\sin\left(\frac{A_{pu}^0}{2\hbar}\sqrt{|\mu_x|^2+|\mu_y|^2}\right)^2 \sin\left(\frac{A_{pr}^0}{2\hbar}\sqrt{|v_x|^2+|v_y|^2}\right)}{(|\mu_x|^2+|\mu_y|^2)(|v_x|^2+|v_y|^2)^{\frac{3}{2}}} \times \\
& -\frac{1}{\tau_{\delta_{y,XX}}} \\
\{ & \frac{1}{\tau_{\delta_{y,XX}}^2} + (\omega_{y,XX} - \omega)^2 \} \left[(|v_y|^2|v_x|^2(|\mu_y|^2 - |\mu_x|^2) + |v_y|^2 \cos\left(\frac{A_{pr}^0}{2\hbar}\sqrt{|v_x|^2+|v_y|^2}\right)(|\mu_y|^2|v_y|^2 + |\mu_x|^2|v_x|^2)) e^{-\frac{t_D}{\tau_r}} + \right. \\
|v_y|^2 \cos\left(\frac{A_{pr}^0}{2\hbar}\sqrt{|v_x|^2+|v_y|^2}\right) & (\mu_y^* \mu_x v_y^* v_x e^{i\omega_y t_D} + \mu_y \mu_x^* v_y v_x^* e^{-i\omega_y t_D}) e^{-\frac{t_D}{\tau_{xy}}} + \\
\frac{|v_x|^2 - |v_y|^2}{2} & (\mu_y^* \mu_x v_y^* v_x e^{i\omega_y t_D} + \mu_y \mu_x^* v_y v_x^* e^{-i\omega_y t_D}) e^{-\frac{t_D}{\tau_{xy}}} \left. + \right. \\
\frac{(\omega_{y,XX} - \omega)}{1 + (\omega_{y,XX} - \omega)^2} & \left. \left[\frac{|v_x|^2 + |v_y|^2}{2i} (\mu_y^* \mu_x v_y^* v_x e^{i\omega_y t_D} - \mu_y \mu_x^* v_y v_x^* e^{-i\omega_y t_D}) e^{-\frac{t_D}{\tau_{xy}}} \right] \right\}
\end{aligned} \tag{R18}$$

and for the x dipole,

$$\begin{aligned}
\Delta A_x^{XX}(\omega, t_D) \sim & -\frac{\sin\left(\frac{A_{pu}^0}{2\hbar}\sqrt{|\mu_x|^2+|\mu_y|^2}\right)^2 \sin\left(\frac{A_{pr}^0}{2\hbar}\sqrt{|v_x|^2+|v_y|^2}\right)}{(|\mu_x|^2+|\mu_y|^2)(|v_x|^2+|v_y|^2)^{\frac{3}{2}}} \times \\
& -\frac{1}{\tau_{\delta_{x,XX}}} \\
\{ & \frac{1}{\tau_{\delta_{x,XX}}^2} + (\omega_{x,XX} - \omega)^2 \} \left[(|v_y|^2|v_x|^2(|\mu_x|^2 - |\mu_y|^2) + |v_x|^2 \cos\left(\frac{A_{pr}^0}{2\hbar}\sqrt{|v_x|^2+|v_y|^2}\right)(|\mu_y|^2|v_y|^2 + |\mu_x|^2|v_x|^2)) e^{-\frac{t_D}{\tau_r}} + \right. \\
|v_x|^2 \cos\left(\frac{A_{pr}^0}{2\hbar}\sqrt{|v_x|^2+|v_y|^2}\right) & (\mu_y^* \mu_x v_y^* v_x e^{i\omega_y t_D} + \mu_y \mu_x^* v_y v_x^* e^{-i\omega_y t_D}) e^{-\frac{t_D}{\tau_{xy}}} + \\
\frac{|v_y|^2 - |v_x|^2}{2} & (\mu_y^* \mu_x v_y^* v_x e^{i\omega_y t_D} + \mu_y \mu_x^* v_y v_x^* e^{-i\omega_y t_D}) e^{-\frac{t_D}{\tau_{xy}}} \left. - \right. \\
\frac{(\omega_{x,XX} - \omega)}{1 + (\omega_{x,XX} - \omega)^2} & \left. \left[\frac{|v_x|^2 + |v_y|^2}{2i} (\mu_y^* \mu_x v_y^* v_x e^{i\omega_y t_D} - \mu_y \mu_x^* v_y v_x^* e^{-i\omega_y t_D}) e^{-\frac{t_D}{\tau_{xy}}} \right] \right\}
\end{aligned} \tag{R19}$$

in which $A_{pu/pr}^0 = \int E_{pu/pr}^0(t)dt$. Given that $\mu_x = \mu_y$, $v_x = v_y$ (with $k_x^{pu/pr} = k_y^{pu/pr}$), and $\mu_y^* \mu_x v_y^* v_x = |\mu|^2 |v|^2 e^{i\varphi}$ with φ being the relative phase between the excitation and readout, and $\Gamma_{deph_XX} = \frac{1}{\tau_{\delta_x, XX}} = \frac{1}{\tau_{\delta_y, XX}}$ defining the linewidth, the TA signal reads that

$$\Delta A_{x/y}^{XX}(\omega, t_D) \sim \frac{\Gamma_{deph_XX}}{\Gamma_{deph_XX}^2 + (\omega_{x/y, XX} - \omega)^2} \left(e^{-\frac{t_D}{\tau_r}} + e^{-\frac{t_D}{\tau_{sy}}} \cos(\omega_{xy} t_D + \varphi) \right) \pm \frac{(\omega_{x/y, XX} - \omega)}{\Gamma_{deph_XX}^2 + (\omega_{x/y, XX} - \omega)^2} e^{-\frac{t_D}{\tau_{sy}}} \sin(\omega_{xy} t_D + \varphi) \quad (R20)$$

Similarly, the TA signal around the exciton resonance can be obtained as

$$\Delta A_{x/y}^X(\omega, t_D) \sim \frac{\Gamma_{deph_X}}{\Gamma_{deph_X}^2 + (\omega_{GS, x/y} - \omega)^2} \left(e^{-\frac{t_D}{\tau_r}} + e^{-\frac{t_D}{\tau_{sy}}} \cos(\omega_{xy} t_D + \varphi) \right) \mp \frac{(\omega_{GS, x/y} - \omega)}{\Gamma_{deph_X}^2 + (\omega_{GS, x/y} - \omega)^2} e^{-\frac{t_D}{\tau_{sy}}} \sin(\omega_{xy} t_D + \varphi) \quad (R21)$$

As for the dynamics around the photoinduced absorption (PIA), we empirically write (because it shares the same excited states)

$$\Delta A_{x/y}^{PIA}(\omega, t_D) \sim \frac{\Gamma_{deph_PIA}}{\Gamma_{deph_PIA}^2 + (\omega_{x/y, PIA} - \omega)^2} \left(e^{-\frac{t_D}{\tau_r}} - e^{-\frac{t_D}{\tau_{sy}}} \cos(\omega_{xy} t_D + \varphi) \right) \mp \frac{(\omega_{x/y, PIA} - \omega)}{\Gamma_{deph_PIA}^2 + (\omega_{x/y, PIA} - \omega)^2} e^{-\frac{t_D}{\tau_{sy}}} \sin(\omega_{xy} t_D + \varphi) \quad (R22)$$

The total TA signal is then calculated by summing all possible contributions that

$$\Delta A \sim \Delta A_x^X + \Delta A_y^X + \Delta A_x^{XX} + \Delta A_y^{XX} + \Delta A_x^{PIA} + \Delta A_y^{PIA} \quad (R23)$$

as shown in Fig. 1d~f in the Main text, which shows an excellent agreement between the experimental and calculated results.

If my understanding is right, CTA is a circular dichroism measurement with circularly polarized pump. In zero field, the (frequency of) oscillations likely measures energy splitting corresponding to exchange couplings, rather than splitting due to anisotropy in crystal field. Furthermore, for the picture of Fig. 1a, a more suitable measurement should be probed by linear dichroism with a linear pump. The author probably showed some data in Fig. S12, mentioning weaker oscillations with same frequency, but there is no further systematic measurements. For both circular dichroism and linear dichroism measurements, it's worth applying magnetic field and directly verify the contribution of Zeeman terms.

We thank the Reviewer for the questions about the linearly polarized measurements. From the derivations as shown above, both circularly and linearly polarized measurements will allow us to measure the quantum beats when the dipoles are simultaneously excited. As shown in Fig. R3, we systematically measured the quantum beats by linearly polarized pump-probe measurements. As mentioned in the Main text, linear polarization-resolved measurements could be done on single NCs with known crystal orientation. For example, to completely remove the beating by perfectly aligning the pump with any dipole. *Since our NCs are randomly orientated, the beats can be observed with any linear polarized pump configuration.* Strictly, the assumptions ($\mu_x = \mu_y, \nu_x = \nu_y$, and hence $k_x^{pu/pr} = k_y^{pu/pr}$) to obtain Eq. R20~21 from Eq. 18~19 is for the case with known dipole orientation. This may not be valid for an ensemble of randomly orientated NCs, which does not allow accurate description of the linearly polarized measurements. However, for circular pump, the electric field is taken as being divided into two components that are perpendicular to each other, which is deemed to more suited for an ensemble of randomly orientated NCs. The spin is optically injected at the same time because of angular momentum conservation along the beam direction. The circular probe is also sensitive to the population of the spin-polarized excitons, which allows us to study the exciton spin relaxation mechanisms. We agree with the Reviewer that it is worth applying magnetic field, however, to measure NCs with determined crystal orientations. Since the main focus of our work is zero-field measurements of the quantum beats and intrinsic mechanisms for spin decoherence, field-dependence experiments could be a future work.

Fig. R3: Quantum beat decay dynamics around the biexciton resonance for NC_{5,1} (see Fig. R1a for TA spectrum) with different linearly polarized pump and probe configurations. (H for horizontal and V for vertical).

Some more details for the authors to check: in the main text, the oscillation period is said to be ~ 4.1 ps, but the period in Fig. 2c is only about 2 ps; it's not clear how to get the corresponding energy of ~ 1.6 meV either.

We thank the Reviewer for pointing this out typo. We have corrected the oscillation period to be ~ 2.5 ps with the corresponding energy of ~ 1.7 meV in Fig. 2c, for sample NC5. We have also checked that all the spin lifetimes and splitting energies estimated carefully. Our main conclusions are not affected by this typo. The corresponding energy is calculated by $\Delta = \hbar\omega_{xy}$ (symbol Ω for the frequency is used in the Main text) with \hbar the reduced Planck constant and ω_{xy} the beating frequency, or by $\Delta = \hbar\frac{2\pi}{T}$ with T the oscillation period. The beating frequency can be obtained by fitting the dynamics using a damped cosine function (with a phase shift) as shown below. The following part has also been added to Supplementary Note. 2:

Set the probing frequency ω to the exciton resonance ($\omega = \omega_{GS,x}, \omega = \omega_{GS,y}$), around the exciton resonance we have

$$\begin{aligned}
S_z \propto (\Delta A_{\sigma^+\sigma^+} - \Delta A_{\sigma^+\sigma^-}) = & \\
& [\Delta A_x^X(\omega_{GS,x}, t_D, \varphi = 0) + \Delta A_y^X(\omega_{GS,y}, t_D, \varphi = 0)] - [\Delta A_x^X(\omega_{GS,x}, t_D, \varphi = \pi) + \Delta A_y^X(\omega_{GS,y}, t_D, \varphi = \pi)] \sim \\
& -\frac{4}{\Gamma_{deph_X}} e^{-\frac{t_D}{\tau_{xy}}} \cos \omega_{xy} t_D
\end{aligned} \tag{R24}$$

Set ω to the biexciton resonance ($\omega = \omega_{x,XX}, \omega = \omega_{y,XX}$), we have the similar result that (because of the selection rule, $\varphi = \pi$ for $\Delta A_{\sigma^+\sigma^+}$ and $\varphi = 0$ for $\Delta A_{\sigma^+\sigma^-}$)

$$\begin{aligned}
S_z \propto (\Delta A_{\sigma^+\sigma^+} - \Delta A_{\sigma^+\sigma^-}) = & \\
& [\Delta A_x^{XX}(\omega_{x,XX}, t_D, \varphi = \pi) + \Delta A_y^{XX}(\omega_{y,XX}, t_D, \varphi = \pi)] - [\Delta A_x^{XX}(\omega_{x,XX}, t_D, \varphi = 0) + \Delta A_y^{XX}(\omega_{y,XX}, t_D, \varphi = 0)] \sim \\
& -\frac{4}{\Gamma_{deph_XX}} e^{-\frac{t_D}{\tau_{xy}}} \cos \omega_{xy} t_D
\end{aligned} \tag{R25}$$

From which the beating frequency ω_{xy} can be obtained.

- 2. The main conclusions of this manuscript include the anomalous temperature dependence of the spin lifetime and explanation of spin relaxation mechanisms in different temperature ranges. The authors have to justify the time scales extracted from fitting the time traces by spin relaxation times, and how the measured dynamics is not dominated by inhomogeneous broadening on ensemble of NCs.*

We thank the Reviewer for raising this question. As mentioned above, the high-temperature spin-relaxation times can be readily obtained by fitting with an exponential decay, which is defined in Supplementary Note 1. At low temperatures, the spin lifetimes can be extracted by fitting with a damped cosine function (Eq. R25) with a phase shift. As-acquired spin lifetimes are then analysed combining with simulation. These fitting procedures do not violate our results because the subtraction between two counter circular probes yields the net spin polarization as defined.

At room temperature, the probe-energy resolved spin lifetimes in Supplementary Fig. S8 shows almost the same spin polarization dynamics. At 12 K, the linewidth of the NC5 ensemble (~ 48 meV) is much smaller than the exciton resonance (~ 2450 meV) which indicates the high sample quality (see also Fig. R1 in this letter). Therefore, we conclude that the size distribution does not affect the spin dynamics much. In addition, from Eq. R24~25 we know that the intrinsic inhomogeneous linewidth in single NC affects only the oscillating amplitude (see also Fig. R1), we conclude that the spin lifetimes are not dominated by the inhomogeneous broadening effects.

Related to both of the points, there is one useful reference, Yugova et al, "Exciton fine structure in InGaAs/GaAs quantum dots revisited by pump-probe Faraday rotation" PRB 75, 195325 (2007).

We thank the Reviewer for the insightful reference on self-assembled InGaAs/GaAs quantum dots by molecular beam epitaxy with known crystal orientations. We agree with the Reviewer that it would be of great interest to investigate the magnetic-field dispersions of exciton fine-structure splitting. Such studies have been done with MAPbBr₃ single crystal (*Nano Lett.*, 19, 10, 7054–7061 (2019)), PEA₂PbI₄ single crystal (*J. Phys. Chem. Lett.*, 13, 20, 4463–4469 (2022)) with known growth direction. Unfortunately, our samples are ensembles of NCs spin-coated onto substrates with random orientations, which does not allow such measurements.

Reviewer #3

Present manuscript reports on experimental studies of lead-halide perovskite, CsPbBr₃, nanocrystals(NCs). The focus is exciton coherent dynamics within the band-edge J=1 exciton triplet whose fine(FSS) structure splitting is attributed to the Rashba effect. The authors use circularly polarized transient absorption (CTA) spectroscopy to probe quantum beats between selected two orthogonally polarized states associated with symmetric and anti-symmetric linear combinations of J_z = -1, 1 exciton states. The authors claim that this is the first reported use of a coherent spectroscopy to directly resolve quantum beats (~4 ps) within the exciton FSS (splitting of 1.6 meV). Subsequent temperature resolved examination of the quantum beat decay time attributed to the spin relaxation time shows a crossover of different spin relaxation dynamics. The authors interpret them to the Elliott-Yafet (EY) at high-T and motional narrowing at low-T. A Mont Carlo simulation results are reported in support of such an interpretation.

Direct observation of the quantum beats within the optically unresolved FSS and experimental identification of different spin-relaxation dynamics as a function of temperature represent strongpoints of this study. However, the manuscript has several drawbacks that the authors need to address.

We thank the Reviewer for the encouraging comments. In the following, we address the drawbacks in our point-by-point response.

Comment 1. The manuscript presentation needs significant improvements. Some statements lack clarity; some are contradictory:

We appreciate the feedback from the Reviewer to help improve our manuscript.

L.110: Reported ~2.39 eV is red shifted bi-exciton energy not the bi-exciton energy detuning as stated. The detuning in this case should be the exciton binding energy not even specified in the manuscript. It'll be useful to know values for the binding energy and make the statement more accurate.

We thank the Reviewer for pointing this out this typo. We have amended this part accordingly and indicated the biexciton binding energy of ~50 meV in the revised manuscript.

Page 4, Line 110-112:

“However, biexciton induced absorption is observed below bandgap with a red-shifted detuning energy of ~2.39 eV for both circular probes as shown in Fig. 2a and 2b” has been changed to “However, biexciton induced absorption is observed below bandgap with a red-shifted energy of ~2.39 eV for both circular probes as shown in Fig. 2a and 2b. The biexciton binding energy is then estimated to be ~50 meV at 12 K”.

LL. 111-113: *The statement that “the photoinduced spectral shift and exciton linewidth changes contribute to the absorption above bandgap” does not explain observed above bandgap absorption but rather raises another equation how those changes contribute to the absorption effect. The authors need to clarify this.*

We thank the Reviewer for the comment. In the original manuscript we wrote that “*the photoinduced spectral shift and exciton linewidth change contribute to absorption above bandgap, for both co-circular and counter-circular geometries despite different intensities at early delay times*”, which was stated to explain the *sub-picosecond* dynamics. As shown in Supplementary Fig. S9 (also Fig. S8 after the subtraction), the photoinduced absorption during the first hundreds of femtoseconds could originate from the optical Stark effect or/and four wave mixing signal. We have amended this part in the revised manuscript to avoid confusion:

Page 4, Line 113-115:

The sentence “*the photoinduced spectral shift and exciton linewidth change contribute to absorption above bandgap, for both co-circular and counter-circular geometries despite different intensities at early delay times*” has been changed to “*the photoinduced spectral shift and exciton linewidth change contribute to absorption above bandgap, for both co-circular and counter-circular geometries despite different intensities at the first few hundred femtoseconds (more clearly in Supplementary Fig. S9)*”.

LL. 126-127: *Like above, why the π -phase shift eliminates the phonon contribution to the exciton coherences?*

We thank the Reviewer for this comment. The lattice vibrations should not be sensitive to the light polarization in halide perovskites (see, for example, *Sci. Adv.* 2, e1600477 (2016)). We have cautioned this part in the revised manuscript:

Page 5, Line 128-130:

The sentence “*Do note that the π -phase shift between the two probes excludes the contribution from phonon coherence*” has been changed to “*We note that the phonon coherence cannot explain the π -phase shift between the two probes because the lattice vibration should not be sensitive to the light helicity in halide perovskites*”.

LL.134-137. *“We note that our measurement provides the average splitting energy between two linearly polarized eigenstates because of the random orientation of exciton’s CoM momentum (in such case P_{ix} and P_{iy} dipole orientation is immaterial)”. This statement is rather confusing. P_{ix} and P_{iy} dipole orientations should be determined by the crystallographic symmetries, the randomization of the CoM momentum averages during the measurements and the response should be consistent with P_{ix} and P_{iy} momentum orientation. The authors need a better explanation of what the “average splitting” means.*

We thank the Reviewer for raising this point. Indeed, the dipole orientations are locked with the crystallographic symmetries. As shown in Fig. R2 (in our response to Reviewer #2), the random orientations of NCs on the substrate average the splitting energy between any two dipoles over the whole ensemble. In our measurements, the z-axis is defined to be fixed and perpendicular to the substrate plane, and hence the dipole orientation is immaterial. The other averaging effect might come from the size distribution, which is shown to be very small as suggested by Supplementary Fig. S8b and Eq. S24~25. We have added this description accordingly in the revised manuscript:

Page 5, Line 137-140:

We note that our measurement provides the average splitting energy between any two linearly polarized eigenstates because of the random orientations (in such case the dipole orientation is immaterial in the substrate plane).

In LL. 215-216, the authors assume time independent Larmor frequency Ω to extract the power-law exponents for momentum relaxation. Below in LL.235-236 they claim that Ω was assumed to be temperature dependant in the MC calculations. These statements contradict to each other.

In L. 258: the intrinsic magnetic field in perovskite NCs “must be momentum dependant”, however, LL. 234-234 make a contradictory statement that in the MC simulations “We assume momentum-independent strength of the effective magnetic field Ω .” This needs to be clarified.

We thank the Reviewer for the comment. The magnitude of the effective magnetic field is assumed to be temperature-independent while being size-dependent. However, the direction of the field is changing upon momentum scattering and hence the temperature-dependence. We apologize for the confusion caused. We have clarified this accordingly in the revised manuscript:

Page 9, Line 222:

“By assuming constant Ω^2 against temperature” has been changed to “By assuming Ω^2 with a constant strength against temperature”.

Page 9, Line 239-244:

“The spin precession is assumed around in-plane momentum-dependent effective magnetic fields originating from the FSS energy splitting, which are changed in direction by spin-preserving, momentum-randomising scatterings. We assume a momentum-independent strength of the effective magnetic field Ω . Yet both Ω and the momentum scattering rate τ_p^{-1} are assumed temperature and NC size dependent and individually fitted to each dataset.” has been changed to

“The spin precession is assumed to be around in-plane momentum-dependent vectors of effective magnetic fields originating from the FSS energy splitting, which changed direction by the spin-preserving, momentum-randomising scatterings. We assume a momentum-independent strength of the effective magnetic field with a constant spin precession frequency of Ω . Yet the momentum scattering rate τ_p^{-1} is assumed to be temperature and NC size dependent and individually fitted to each dataset.”

LL.248-249 state that the exciton spin relaxation dynamics can be divided into three regimes. From the discussion below it is hard to say what those three regimes are. It'll be helpful for the reader if these three regimes are clearly itemized (in addition see related Comment 4).

We thank the Reviewer for this feedback. The three regimes for exciton spin relaxation are schematically summarized in Fig. 3b. We have itemized these three regimes accordingly in the revised manuscript. The following description has been added in the revised manuscript at the beginning of the discussion section:

Page 10, Line 257-262:

From high to low temperature, the exciton spin relaxation in CsPbBr₃ NCs can be divided into three regimes with different dominant mechanisms: 1) the EY mechanism where the scattering with LO phonons depolarizes spins; 2) the strong-scattered motional narrowing mechanism where the momentum scattering preserves spin polarization; and 3) the quasi-scattering-free motional narrowing of spins which yields oscillatory spin dynamics. These as-described exciton spin relaxation processes are schematically shown in the insets of Fig. 3b.

LL 279-280. “The large bi-exciton binding energy help the coherence survive at high temperature”. This concluding statement seems totally misaligned with the manuscript main theme focused on the spin relaxation processes due to the exciton centre of mass motion. The electron-hole exchange processes are contributing to both the exciton/bi-exciton binding energies and to the spin relaxation dynamics. However, the manuscript has no clear study of these contributions not even a discussion about the relationship between the X/XX binding energies and relaxation mechanisms.

We thank the Reviewer for the feedback. We agree with the Reviewer that the exciton/biexciton binding energy is out the scope of our work. We have removed this statement in the revised manuscript.

Supplementary Information Eq. S4 contains a sum of different contributions to the CTA signal, however, only bi-exciton absorption term is given in Eq. S3. The authors should list the rest of the terms.

We thank the Reviewer for the feedback. We have provided more details about the derivation and listed all relevant terms for the CTA signal in the revised Supplementary Information, which are also shown by Eq. R20~23 in this letter.

Comment 2: The interpretation of the quantum beat decay is solely given based on the spin-flip, i.e., population decay T1-dynamics. This justifies the rate equations S1 and S2 used by the authors. However, pure-dephasing T2 mechanisms can be involved into the quantum beats decay which is common to consider in the pump-probe spectroscopies. In the NMR and indeed in visible/optical range such an interplay is accounted for by describing the dynamics with the help of the Redfield equation. The authors need to provide solid arguments towards eliminating the pure dephasing mechanisms of the quantum beat decay from consideration.

We thank the Reviewer for the insightful comment. We have taken the pure dephasing terms into consideration to derive the TA signal in the revised Supplementary Information (also shown in the response to the Reviewer #2 in this letter), where we use the Lindblad dissipator in the quantum mechanical Liouville equation to describe the relaxation and dephasing processes. As shown by Eq. R24~25, the interband pure dephasing time affects only the oscillating amplitude. The pure dephasing time between two FSS states does affect the quantum beat decay time, which yields a coherence time τ_{xy} that is much shorter than the population lifetime.

Comment 3: Bringing out linearly polarized line measurements in LL. 137-145 raises questions of why the quantum beat decay shows down, what relaxation mechanisms are involved and how this is aligned with the CTA signal interpretation. The questions are left open and never revisited in the manuscript.

We thank the Reviewer for the comment on linearly polarized measurements. The CTA signals are given by general equations for arbitrary pump and probe polarizations as shown in the response to the Reviewer #2 in this letter (Eq. R18~19). The observation of quantum beating between FSS states is available when the FSS states are simultaneously excited, either by linearly polarized pump that is not aligned perfectly to one dipole, or by circularly polarized pump. Since our NCs are randomly orientated on the substrate, there are always simultaneously excited dipoles in some NCs for any linear polarization pump (Fig. R3). However, the assumptions ($\mu_x = \mu_y$, $\nu_x = \nu_y$, and hence $k_x^{pu/pr} = k_y^{pu/pr}$) to obtain Eq. R20~21 from Eq.

18~19 may not be valid for an ensemble of randomly orientated NCs, which do not allow accurate description of the linearly polarized measurements. Linear-polarization-resolved studies can be done with single NCs as suggested in the Main text, to measure the transverse relaxation time (in substrate plane).

Comment 4: In the discussion of possible motional narrowing mechanisms, the authors refer to the D'yakonov-Perel' (DP) and Maialle-Silva-Sham (MSS) mechanisms. The latter seems to be important and is brought into consideration in analogy to detailed study of 2D lead-halide perovskites nanoplates (NPs) Ref. 35. That study concludes that the dynamic polaritonic screening facilitates the MSS mechanism in the NPs. In contrast, in the NCs the exciton-polariton effects are suppressed (see, e.g., Section SIG in Ref. 23). It is reasonable to expect that the Rashba effect could be a contributor to the MSS. This aspect has not been studied in the NCs and, therefore, the manuscript lacks detailed examination of the MSS mechanism and comparison with the results for 2D NPs, Ref. 23. If the authors provide such an examination, the impact of the paper will be enhanced to the level required by the Nature Communication.

We thank the Reviewer for the comment on possible spin relaxation mechanisms in perovskite NCs. Both DP and MSS mechanisms can be categorized, in the language of NMR, as the motional narrowing of carrier/exciton spins. To the best of our knowledge, the MSS mechanism has *only* been considered in quantum wells (*Phys. Rev. B Condens. Matter* 47, 15776-15788 (1993), *Phys. Rev. B Condens. Matter* 51, 4247-4257 (1995), *Nano Lett.* 18, 223-228 (2018)), because the one-dimensional spatial confinement and the out-of-plane exciton spins lose coherence due to the in-plane exchange field. For orthorhombic nanocrystals, asymmetry may exist along all three dimensions while the excitons are not strictly spatially confined. According to Ref. 23, the possible existence of Rashba Hamiltonian further breaks the symmetry, acting on the FSS states. As a result, the Rashba effect is responsible for the exciton fine structure. Therefore, such “*similar MSS mechanism*” has already been taken into consideration in our modelling as a combined effect when we interpret the spin decoherence with the effective magnetic field induced by FSS, and we carefully used the general term “*motional narrowing*” to describe it. As for polaron, it is more related with the momentum scattering mechanisms, which indirectly affects the temperature-dependent behaviour of spin relaxation in the LO phonon-dominated regime (manifest as the power exponent k in the Main text). The temperature dependence of the spin lifetime shows a similar trend with that in Ref. 35, while our low-temperature oscillatory spin dynamics fingerprints the motional narrowing process in a quasi-scattering-free regime.

REVIEWERS' COMMENTS

Reviewer #1 (Remarks to the Author):

My concerns have been addressed properly in this revised version, and I would like to recommend the current version for publication.

Reviewer #3 (Remarks to the Author):

The authors have provided an extensive and satisfactory response to my comments and critiques. Their revision of the manuscript significantly improved the presentation. To my opinion the experimental results are new, and their theoretical interpretation is adequate. Accordingly, I would recommend the manuscript publication in Nature Communication as it is.

Reviewer #4 (Remarks to the Author):

The manuscript provided by Cai et al., reports on experimental observation of excitonic quantum coherence and identifies different exciton spin relaxation mechanisms in CsPbBr₃ nanocrystals, and both are supported by theory. After reading and thinking the previous Reviewers' comments carefully, I completely agree with the Reviewers that it is of great interest and can make an important contribution to the community. The authors provided a well-organized response to the Reviewers' comments and addressed most of the concerns with additional inputs. The only remaining issue might be the magnetic-field dependent study. Nonetheless, as indicated, the field control of the exciton quantum beats is challenging in ensemble of randomly distributed nanocrystals, and might be out of the scope of this work. I encourage the authors to continue this research at single-particle level.

Therefore, I support this manuscript for publication on Nature Communications, after addressing the following minor revisions,

1. Since CsPbBr₃ is the only studied halide perovskite in this manuscript, it would be better to refine the title by "... in CsPbBr₃ nanocrystals". It is not accurate to draw conclusions for all halide perovskite nanocrystals simply, because there're organic-inorganics perovskite nanocrystals (MAPbX, FAPbX, ...) that are quite different from their all-inorganic counterparts.
2. As reported in the manuscript, the nanocrystals show extremely short exciton spin lifetimes down to sub-picosecond. What is the IRF of the setup used and how does it affect the estimate of the lifetime?
3. The authors claimed "average splitting" measured, how to evaluate the average splitting, measured by circularly polarized transient absorption spectroscopy, different from single-particle PL measurements (Nature 553 (2018) 189–193)? Can the authors comment on these techniques?
4. The authors have already shown some linearly-polarized measurements in their response, why not add some into the supplementary file to give readers more clues (see fig. S12)?
5. There're several review papers (Adv. Optical Mater. 7 (2019) 1900350; Advanced Optical Materials, 9(2021) 2100215; Advanced Quantum Technologies 4 (2021) 2100052) regarding on the topic observed in this manuscript, the reviewer suggest the authors should read them carefully and cite them in proper position in the Introduction, to provide an overall review for readers.

Response to the Reviewers

We would like to thank all the Reviewers again for their feedback. We have also made our best efforts to address the additional concerns and revise the manuscript. In this letter, comments from reviewers are shown in *green italics*, our responses are given in black, our highlighted revisions are given in *blue italics*, and deleted sections from the original manuscript are given in *red italics*.

Reviewer #1

My concerns have been addressed properly in this revised version, and I would like to recommend the current version for publication.

We are delighted by the reviewer's recommendation for publication. We thank the Reviewer again for his valuable time to help us strengthen our work.

Reviewer #2

The authors have provided an extensive and satisfactory response to my comments and critiques. Their revision of the manuscript significantly improved the presentation. To my opinion the experimental results are new, and their theoretical interpretation is adequate. Accordingly, I would recommend the manuscript publication in Nature Communication as it is.

We are delighted to hear that our revision has addressed the concerns and improved the data presentation. We thank the Reviewer again for the insightful comments, which helped us to improve the manuscript.

Reviewer #4

The manuscript provided by Cai et al., reports on experimental observation of excitonic quantum coherence and identifies different exciton spin relaxation mechanisms in CsPbBr₃ nanocrystals, and both are supported by theory. After reading and thinking the previous Reviewers' comments carefully, I completely agree with the Reviewers that it is of great interest and can make an important contribution to the community. The authors provided a well-organized response to the Reviewers' comments and addressed most of the concerns with additional inputs. The only remaining issue might be the magnetic-field dependent study. Nonetheless, as indicated, the field control of the exciton quantum beats is challenging in ensemble of randomly distributed nanocrystals, and might be out of the scope of this work. I encourage the authors to continue this research at single-particle level.

Therefore, I support this manuscript for publication on Nature Communications, after addressing the following minor revisions,

We thank the Reviewer for the positive comments on our work and the support for publication. Our point-to-point response to the comments is as follows:

1. Since CsPbBr₃ is the only studied halide perovskite in this manuscript, it would be better to refine the title by "... in CsPbBr₃ nanocrystals". It is not accurate to draw conclusions for all halide perovskite nanocrystals simply, because there're organic-inorganics perovskite nanocrystals (MAPbX, FAPbX, ...) that are quite difficult from their all-inorganic counterparts.

We agree with the Reviewer that it might not be exact to draw the same conclusions for organic-inorganic perovskite nanocrystals. We have amended the title from “*Zero-Field Quantum Beats and Spin Decoherence Mechanisms in Halide Perovskite Nanocrystals*” to “*Zero-Field Quantum Beats and Spin Decoherence Mechanisms in CsPbBr₃ Perovskite Nanocrystals*”.

2. As reported in the manuscript, the nanocrystals show extremely short exciton spin lifetimes down to sub-picosecond. What is the IRF of the setup used and how does it affect the estimate of the lifetime?

We thank the Reviewer for the comment on the IRF. The pump-probe cross-correlation is ~130 fs in our setup, which has been considered in the fittings to extract the exciton spin lifetimes (> ~600 fs) for our samples (Fig. 2j in the Main text). Therefore, our analysis is not affected by the IRF of the setup.

3. The authors claimed "average splitting" measured, how to evaluate the average splitting, measured by circularly polarized transient absorption spectroscopy, different from single-particle PL measurements (Nature 553 (2018) 189–193)? Can the authors comment on these techniques?

We thank the Reviewer for raising this point about the measurements of exciton fine structure splitting (FSS) in CsPbBr₃ nanocrystals. We have added relevant comments in the Discussion section.

Page 9, Line 248-255:

In addition, we demonstrate that ultrafast spectroscopy allows estimating the exciton fine-structure splitting in ensembles of colloidal perovskite nanocrystals at zero magnetic field, even though that the exciton multi-levels are not spectrally resolved. However, such measurements can only provide average splitting because of the random dipole distribution and nanocrystal size distribution. Furthermore, the quantum beats may be smoothed with increasing

temperature due to the motional narrowing process. On the other hand, single-particle PL measurements allow direct observation of the FSS levels with high energy resolution.

4. The authors have already shown some linearly-polarized measurements in their response, why not add some into the supplementary file to give readers more clues (see fig. S12)?

We thank the Reviewer for pointing out this issue. We have updated the results of linearly-polarized measurements in the revised Supplementary Information.

Supplementary Information, Fig. S12:

5. There're several review papers (Adv. Optical Mater. 7 (2019) 1900350; Advanced Optical Materials, 9(2021) 2100215; Advanced Quantum Technologies 4 (2021) 2100052) regarding on the topic observed in this manuscript, the reviewer suggest the authors should read them carefully and cite them in proper position in the Introduction, to provide an overall review for readers.

We thank the Reviewer for the references. We have now included them in the revised manuscript (refs 22, 23, and 39) to give a broader overview for readers.